# The downscaling and adjustment method ADAMONT v1.0 for climate projections in mountainous regions applicable to energy balance land surface models

Deborah Verfaillie<sup>1</sup>, Michel Déqué<sup>1</sup>, Samuel Morin<sup>1</sup>, and Matthieu Lafaysse<sup>1</sup> <sup>1</sup>CNRM UMR 3589, Météo-France/CNRS, Toulouse, France *Correspondence to:* Deborah Verfaillie (deborah.verfaillie@meteo.fr)

# Abstract.

We introduce a method - called ADAMONT (ADAptation of RCM outputs to MOuNTain regions) v1.0 - to downscale and adjust daily climate projections from a regional climate model against a regional reanalysis of hourly meteorological conditions using quantile mapping. The method pro-

- duces adjusted hourly time series of temperature, precipitation, wind speed, humidity, and short- and longwave radiation, which can in turn be used to force any energy balance land surface model. The ADAMONT method is evaluated through its application to the ALADIN-Climate v5 RCM forced by the ERA-Interim reanalysis, compared to the SAFRAN reanalysis, used as the pseudo-observation database covering the entire French Alps split into 23 massifs within which meteorological con-
- ditions are provided for several elevation bands separated by 300 m altitude. Different evaluation criteria are analysed for temperature, precipitation, but also snow depth, which is computed by the SURFEX/ISBA-Crocus model using the meteorological driving data generated using this method. The impact of the learning period and of the method used to select neighbouring RCM grid points for each SAFRAN massif/altitude configuration is tested. The performance of the method is satisfying,
- with similar or even better evaluation metrics than previous literature findings. Results for temperature are generally better than for precipitation. Snow depth yields good results, which can be viewed as indicating a reasonably good inter-variable consistency of the meteorological data produced by the method. The temporal transferability of the method is assessed through the comparison of results obtained using different learning periods, and shows that the method is sensitive to the period con-
- sidered due to the empirical treatment of values beyond the 99.5<sup>th</sup> quantile. The use of a complex RCM grid points selection technique taking into account horizontal but also altitudinal proximity to SAFRAN massif centroids/altitude couples generally degrades evaluation metrics for high altitudes, compared to a simpler 2-dimensional proximity selection technique.

#### 1 Introduction

- Projections of future climate change have been increasingly considered in recent years, as the reality of climate change has been gradually accepted and societies and governments have started to plan upcoming mitigation and adaptation policies (IPCC, 2013, 2014a, b, c). For a given socio-economic or greenhouse-gas concentration scenario, these projections generally concern future temperature and precipitation, and associated extreme events, and are usually achieved using the outputs of gen-
- eral circulation models (GCMs) downscaled using statistical or dynamical methods (e.g. Giorgi and Mearns, 1991; Fowler et al., 2007; Themeßl et al., 2012). In mountain regions such as the Alps, where a large fraction of the income comes from winter tourism (about 20% of the French tourism income in 2011, DSF, 2011) and water and hydro-power production, particular attention is brought to current and future snow availability (e.g. Martin et al., 1994; Beniston, 1997; Castebrunet et al.,
- 2014; Piazza et al., 2014; Schmucki et al., 2014). The question of the associated changes in river discharges (Lafaysse et al., 2014), their consequences on water storage management (François et al., 2015), the future vulnerability of alpine ecosystems (Boulangeat et al., 2014; Thuiller et al., 2014) as well as the future occurrence of climate-related hazards such as debris flows (Jomelli et al., 2009) and avalanches (Castebrunet et al., 2014) is also under consideration. In order to generate projections of
- snow conditions, however, using operational regional climate models (RCMs) featuring spatial resolutions typically between 10 and 50 km, driven by GCMs in a dynamical downscaling approach is not sufficient to capture the fine-scale processes and thresholds at play. Indeed, the altitudinal resolution matters, since the precipitation phase is mainly controlled by the temperature which is altitude-dependent. Several mountain socio-economic activities, such as ski resorts operations, crit-
- ically depend on the altitude of the rain/snow transition. Furthermore, in addition to their lack of representation of the entire altitude range of mountainous areas, simulations from GCMs and RCMs suffer from biases compared to local observations (Christensen et al., 2008; Rauscher et al., 2010; Kotlarski et al., 2014). Raw climate projections must therefore be adjusted and further downscaled (Déqué, 2007; Themeßl et al., 2011; Gobiet et al., 2015), before they can be used as such (time series of temperature, precipitation, ...), or in order to force specific impact models.

Various downscaling and bias adjustment methods have been developed (Maraun et al., 2010; Teutschbein and Seibert, 2012, 2013). They are generally separated into two categories depending on the approach used, but they all require an observation dataset which (i) meets the data requirements of the impact model and (ii) is sufficiently long and reliable to be used to infer the relationships

- between the observations and the raw climate projections during the observation time period. Perfect prognosis approaches search for relationships between observed large-scale predictors (generally from reanalyses) and observed local-scale predictands. The analog (Dayon et al., 2015) and the weather typing (Vrac et al., 2007a) methods fall into this category. In contrast, model output statistics approaches calibrate model outputs against observations. Different methods of variable complexity
- belong to this category, such as scaling methods (linear scaling, local intensity scaling, variance scal-

ing, ...), delta-change methods (e.g. Abegg et al. (2007); Hantel and Hirtl-Wielke (2007); Schmucki et al. (2014)) and distribution mapping methods (e.g. Boé et al. (2007); Déqué (2007); Gobiet et al. (2015); Olsson et al. (2015)). The latter include quantile mapping, which is considered as an efficient and easy to implement adjustment method (Themeßl et al., 2011; Teutschbein and Seibert,

- 2012; Maurer and Pierce, 2014; Gobiet et al., 2015). The main advantage of this method is that it adjusts deviations in the shape of the distribution, and is thus able to adjust deviations not only for the mean but the variability (Themeßl et al., 2011). Moreover, the adjustment is not strictly restricted to the range of observed values in the reference period, which is the case for example for methods based on analog weather patterns (e.g. Déqué, 2007; Themeßl et al., 2011; Rousselot et al., 2012; Dayon
- et al., 2015). It can thus be used for evaluation of climate extremes or projections at the end of the 21<sup>st</sup> century, as long as the probability associated to these events is robustly estimated from a long enough sample. The main limits of quantile mapping are the assumption of time-invariant model biases on which it is based (which is an assumption common to most bias adjustment methods), and the fact that temporal properties of the model are not adjusted, i.e. if the model has a chronolog-
- ical behaviour which differs from the observations (too chaotic or too persistent), this will not be adjusted (Déqué, 2007). Moreover, quantile mapping does not guarantee spatial and inter-variables consistency, which is the case for the analog method for example.

The choice of the downscaling and adjustment method implemented depends on the application and on the availability of raw climate projections and appropriate observation datasets. Raw

- regional climate projection data are increasingly made available to a broad scientific community, e.g. the World Climate Research Program (WCRP) Coordinated Regional Downscaling Experiment (CORDEX, Giorgi et al. (2009)), which aim is to improve and distribute regional climate modelling worldwide. Its European branch, EURO-CORDEX (Jacob et al., 2014), gathers regional climate simulations over Europe from 30 different modelling groups at 50 km (EUR-44) and 12.5 km (EUR-11)
- resolution. On the observation side, the use of surface meteorological reanalysis is a powerful alternative to station observation data to provide the necessary observational dataset (Berg et al., 2015). Indeed, the process by which such reanalyses are generated address the time and space variations of the meteorological conditions, and by design they consist of gap-free and complete time series. In mountainous regions, snow on the ground plays a pivotal role, but the snowpack response to mete-
- orological conditions depends on intertwined processes, involving surface mass and energy balance as well as internal processes (e.g. Martin et al., 1994). Not only temperature and precipitation act on the snowpack, but a broader range of meteorological conditions and their diurnal variations. As a consequence, considering only downscaled and adjusted daily temperature and precipitation would miss some of the non-linear response of the snowpack. The SAFRAN meteorological analysis has
- been developed specifically to address the needs of snowpack numerical simulations, and contains hourly time series of temperature, precipitation, wind speed, humidity, and short- and longwave radiation for so-called massifs (ranging between 500 and 2000 km<sup>2</sup> in the French Alps) by elevation

steps of 300 m (Durand et al., 2009a, b). Combining the strength and efficiency of the quantile mapping downscaling and adjustment methods using the SAFRAN reanalysis as a pseudo-observation

- dataset in order to drive energy balance snowpack and land surface models is a highly desirable goal making full use of the current capabilities of climate impact assessment tools for mountainous regions. In addition, the use of such methods ensures that the chronology of the adjusted and downscaled climate projections will match the chronology of the RCM, which may be affected by climate change through variations of the seasonality of meteorological conditions. Such impacts
- cannot be addressed using delta-change methods, which by definition apply fixed changes to an observed time series conserving its statistical persistence properties and seasonality (e.g. Abegg et al., 2007; Hantel and Hirtl-Wielke, 2007; Schmucki et al., 2014) although this could evolve significantly under changed climate conditions.

In this study we introduce a method - the ADAMONT (ADAptation of RCM outputs to MOuN-

- Tain regions) v1.0 method to downscale and adjust biases in climate model projections in order to provide hourly adjusted temperature, precipitation, wind speed, humidity, and short- and longwave radiation for recent and future time series. Quantile mapping is applied for all these variables using daily outputs from a RCM, which are downscaled and statistically adjusted against the SAFRAN reanalysis (Durand et al., 2009b), an extensive reanalysis of these various meteorological variables
- which operates at the massif scale by 300 m elevation bands. These adjusted fields are then used to force the SURFEX/ISBA-Crocus (Vionnet et al., 2012) model over the French Alps. In order to evaluate the performance of the ADAMONT method, here we apply this method to the ALADIN-Climate v5 RCM (Colin et al., 2010) forced by the ERA-Interim reanalysis (Dee et al., 2011). Sect. 2 focuses on the models used and the evaluation of the ADAMONT method. The performance of the

method is then presented and discussed in Sect. 3, and general conclusions are drawn in Sect. 5.

#### 2 Models and methods

#### 2.1 RCM simulation

In this study, our downscaling and adjustment method is applied to the Météo France RCM AL-ADIN. This RCM, forced by the ERA-Interim reanalysis, was run daily at 12.5 km resolution. This

- reference simulation was then downscaled and adjusted against the SAFRAN reanalysis (Sect. 2.2) using the method described in Sect. 2.4. We chose to work on a spatial domain smaller than the one used in EURO-CORDEX (domain covering all of Europe), in order to evaluate the method and not the output of the RCM itself. Indeed, the smaller the domain, the more it is constrained (Alexandru et al., 2007). Simulations carried out over a domain centered on France, called FRB12 (Fig. 1) were
- thus evaluated (Sect. 2.5).

#### 2.2 SAFRAN reanalysis

The SAFRAN system is a meteorological downscaling and surface analysis system (Durand et al., 1993), which provides hourly data of temperature, precipitation amount and phase, specific humidity, wind speed, and shortwave and longwave radiation for each mountain region (called massif) in the

- French Alps (23 massifs, as illustrated in Fig. 1). Massifs (Durand et al., 1993, 1999) correspond to regions ranging approximately between 500 and 2000 km<sup>2</sup> for which meteorological conditions are assumed to be spatially homogeneous but vary with altitude. SAFRAN data are available for elevation bands with a resolution of 300 m. SAFRAN was used by Durand et al. (2009b) to create a meteorological reanalysis over the French Alps by combining the ERA-40 reanalysis (Uppala et al.,
- 2005) with various meteorological observations including in situ mountain stations, radiosondes and satellite data. It was complemented after the end of the ERA-40 reanalysis (2002) by large-scale meteorological fields from the ARPEGE analysis, so that it now spans the period from 1959 to 2015, making it one of the longest meteorological reanalyses available in the French mountain regions.

# 2.3 SURFEX/ISBA-Crocus model

- Crocus (Brun et al., 1989, 1992) is a detailed snowpack model which has been used operationally for French avalanche forecasting for more than 20 years. It has been recently integrated in the SURFEX externalised surface module (Masson et al., 2013), which contains the land surface scheme ISBA (Interactions between Soil, Biosphere, and Atmosphere). The new SURFEX/ISBA-Crocus model (Vionnet et al., 2012) enables the computation of the exchanges of energy and mass between the
- snow surface and the atmosphere (radiative balance, turbulent heat and moisture fluxes, ...), but also between the snowpack and the ground underneath. The one-dimensional multilayer physical snow scheme Crocus is able to simulate the evolution of the snowpack over time, by accounting for several processes occurring in the snowpack, such as thermal diffusion, phase changes, metamorphism, etc. A detailed description of the processes taken into account and the variables of the Crocus scheme can
- be found in Brun et al. (1989, 1992) and Vionnet et al. (2012). The SAFRAN-Crocus model chain has been operationally used for more than 20 years for avalanche hazard forecasting and extensively evaluated over the alpine domain in particular against snow depth observation stations (Durand et al., 1999, 2009b; Lafaysse et al., 2013).

#### 2.4 Description of the ADAMONT method

RCM outputs fed by GCMs suffer from systematic deviations compared to local observations, which are partly due to the model configuration and boundary conditions provided by the GCM, but are also linked to their insufficient spatial and altitudinal resolution (Déqué and Somot, 2008; Kotlarski et al., 2012, 2014). The spatial and altitudinal differences between the ARPEGE/ALADIN simulations compared to the SAFRAN reanalysis are illustrated in Fig. 1. It can clearly be seen that the model

- grid points closest to the SAFRAN massifs centroids are not located directly above the centroids (potential spatial bias), but more importantly, have a surface elevation which can differ significantly from the different altitudes found in each massif (potential altitudinal bias). To mitigate such biases compared to observations, bias adjustment methods are often used before running specific models such as energy balance land surface models.
- The adjustment method used in this study is the quantile mapping method (Déqué, 2007; Gobiet et al., 2015). This method, which is considered one of the most efficient bias adjustment methods available (Themeßl et al., 2011; Maurer and Pierce, 2014; Gobiet et al., 2015) is relatively easy to implement. It consists in plotting the quantiles of the simulated historical distribution against the quantiles of the observed distribution (quantile-quantile plot), and using the resulting mapping
- function to adjust the distribution of model projections. Because the spatial resolution of the adjusted RCM is close to the resolution of the observations (SAFRAN massifs), there is no risk of introducing any artificial inflation of the simulated series, and therefore quantile mapping is adapted (Maraun, 2013). As mentioned above, no bias-adjustment method is perfect, and quantile mapping suffers from some disadvantages, namely the assumption of time-invariant model biases, the fact that
- temporal properties of the model are not corrected for and the fact that the spatial and inter-variables consistency is not guaranteed. Driouech et al. (2009) showed that for zonal climates, such as the one of Morocco, quantile mapping adjustment can vary for different weather regimes. Similarly, Addor et al. (2016) demonstrated the sensitivity of quantile mapping adjustment to circulation biases over the Alpine domain. To limit the dependence of our method to climate, weather regimes are thus taken 185 into account in our method.
- 165 Into account in our method.

The downscaling and adjustment method developed in this study consists in the following steps:

 The RCM grid points closest (in terms of horizontal distance) to the SAFRAN massifs centroids are selected, and the surface data (temperature, total precipitation, wind speed, specific humidity and incoming short- and longwave radiation) of each selected grid point are applied directly to the different elevations of the corresponding SAFRAN massif. An alternative was tested, where the RCM grid points closest to the SAFRAN massifs centroids in terms of horizontal AND vertical distances are selected, by minimizing the distance :

$$\sqrt{(\Delta x)^2 + (\Delta y)^2 + (N \times \Delta z)^2},\tag{1}$$

where  $\Delta x$ ,  $\Delta y$  and  $\Delta z$  represent the longitudinal, latitudinal and vertical distances (in km) between SAFRAN massifs centroids and the RCM grid points, and N is referred to as the elevation factor. The latter was introduced to take into account the strong dependence of meteorological variables (mainly precipitation and temperature) on altitude (e.g. Gottardi et al., 2012; Kotlarski et al., 2012).

 Four different daily weather regimes were diagnosed from ERA-Interim, based on the geopotential height at 500 hPa, following Michelangeli et al. (1995).

- 3. SAFRAN data are integrated from hourly to daily to match the data content of the available RCM output. The integration method depends on the variable considered (see Table 1).
- 4. Percentiles of the RCM distribution and the SAFRAN distribution are then calculated at each point, for each variable, each season (DJF, MAM, JJA, SON) and each weather regime. Each snow year is considered from the 1<sup>st</sup> of August to the 31<sup>st</sup> of July of the following year.
- 5. Quantile mapping is then applied to the entire RCM dataset for the period 1980-2010, taking into account the season and the weather regime.
- 6. For each day in the RCM dataset, an analogous date is chosen in the SAFRAN dataset, matching the following criteria: the month and the regime must be the same as in the RCM dataset, mean precipitation over the Alps must be consistent between datasets (i.e. if there is no precipitation in the RCM, precipitation in the SAFRAN analogue must be zero), and whenever possible, consecutive time slices are chosen in the SAFRAN dataset in order to avoid artificial jumps in the final data linked to the choice of analogues.
- 7. The adjusted RCM dataset is then disaggregated from a daily integration period into an hourly
  time step, by using the hourly SAFRAN data from each analogous date chosen in the previous
  step to reconstruct the daily cycle of the data. Different disaggregation criteria (mean daily
  value, maximum and minimum values, value of the last time step, smallest possible jump between consecutive days, ...) are chosen depending on the variable considered (Table 1). For the
  disaggregation of ALADIN adjusted temperature from daily to hourly (Table 1), a compromise
  must be made between obtaining minimum and maximum daily values as close as possible to
  ALADIN adjusted daily minimum and maximum and minimizing the possible jump in adjusted values between consecutive days. This is achieved by minimising the function:

$$Q(\alpha) = [T_{AL}^{h}(1h,i) - T_{AL}^{h}(24h,i-1)]^{2} + \alpha [Tmin_{AL}^{h}(i) - Tmin_{AL}^{d,adj}(i)]^{2} + \alpha [Tmax_{AL}^{h}(i) - Tmax_{AL}^{d,adj}(i)]^{2},$$
(2)

where  $T_{AL}^{h}(1h,i)$  and  $T_{AL}^{h}(24h,i-1)$  are the hourly adjusted ALADIN temperature values at the first time step of day i and at the last time step of day i-1,  $Tmin_{AL}^{h}(i)$  and  $Tmax_{AL}^{h}(i)$ are the hourly minimum and maximum adjusted ALADIN temperature values respectively, and  $Tmin_{AL}^{d,adj}(i)$  and  $Tmax_{AL}^{d,adj}(i)$  are the daily minimum and maximum adjusted AL-ADIN temperature values respectively (Fig. 2).  $\alpha$  is a parameter which can be tuned to balance the importance of the minimisation of differences between daily and hourly ALADIN minima and maxima and the minimisation of the jump between two consecutive days. For a value of  $\alpha$  of zero, there would be no jump in values between consecutive days, but the values of  $Tmin_{AL}^{h}(i)$  and  $Tmax_{AL}^{h}(i)$  would be far from the values of  $Tmin_{AL}^{d,adj}(i)$  and  $Tmax_{AL}^{d,adj}(i)$ . For an infinitely big value of  $\alpha$ , on the opposite, minimum and maximum

hourly and daily values would match, but the jump between consecutive days could be important. Sensitivity tests yielded an optimal value of 2 for  $\alpha$ . Hourly adjusted ALADIN temperature can be expressed as a function of (hourly) SAFRAN temperature, as:

$$T^h_{AL}(i) = a \times T^h_{SAF} + b, \tag{3}$$

where  $T_{AL}^{h}(i)$  is the hourly adjusted ALADIN temperature and  $T_{SAF}^{h}$  is the hourly SAFRAN temperature from the chosen analogous date (step 6). As a result, eq. 2 transforms into:

$$Q(\alpha, a, b) = [a \times T_{SAF}^{h}(1h) + b - T_{AL}^{h}(24h, i - 1)]^{2} + \alpha [a \times Tmin_{SAF}^{h} + b - Tmin_{AL}^{d,adj}(i)]^{2} + \alpha [a \times Tmax_{SAF}^{h} + b - Tmax_{AL}^{d,adj}(i)]^{2}.$$
(4)

By searching for the local minima  $\delta Q/\delta a = 0$  and  $\delta Q/\delta b = 0$ , *a* and *b* can be determined, and the hourly adjusted ALADIN temperature can be obtained following eq. 3. This procedure is only applied for temperature, because the use of the maximum and minimum criterion can lead to important jumps between consecutive days, which is not the case for other variables (Table 1).

8. Finally, total precipitation is separated into rainfall and snowfall based on hourly adjusted temperature (a threshold of 1 °C is used for the transition from snow to rain, consistent with the approach used in SAFRAN). As mentioned before, inter-variable consistency is not guaranteed by quantile mapping. Consistency between temperature and precipitation is the most critical in this study, because we focus on mountain regions where snow plays an important role. As precipitation and temperature were corrected independently from each other (step 5), the relationship between temperature and precipitation may be modified by quantile mapping, so that the RCM rain and snow distributions may lose consistency. To avoid this, Olsson et al. (2015) separate their temperature data into wet and dry days before adjustment. In our case an additional quantile mapping against SAFRAN is applied for daily cumulated RCM rainfall and snowfall separately. Hourly adjusted RCM rainfall and snowfall are then determined as for total precipitation similar to step 7.

### 2.5 Method evaluation

To evaluate our downscaling and bias adjustment method, adjusted outputs from the ALADIN RCM forced by ERA-Interim (1979-2010) were analysed. The evaluation was carried out for temperature and total precipitation, but also for the snow depth, which integrates all the meteorological variables considered in the downscaling and adjustment method. This allows evaluating the ability of our method to correctly represent integrated outputs computed using SURFEX/ISBA-Crocus from meteorological variables adjusted independently from each other. This is also usually done with river

235

8

discharge for downscaling methods used for hydrological applications (e.g. Lafaysse et al. (2014); Olsson et al. (2015)).

Different criteria were evaluated for each massif of the French Alps (Fig. 1), at every elevation band, and aggregated at the scale of the Northern French Alps (14 massifs) and the Southern French Alps (9 massifs), which represent regions with distinct climates (Durand et al., 2009b). Massifs are

270 grouped in the Southern and Northern French Alps by calculating the mean value of temperature, precipitation or snow depth for a given altitude, which is then evaluated with the different criteria. For specific scores concerning the detection and the persistence of precipitation events (listed below), scores are first determined for each massif, and then an average score is calculated for each altitude for the different massifs of the Northern and the Southern French Alps. Some evaluations

- are presented for all altitudes in Sect. 3, others only for 1200 m a.s.l. (above sea level) and 2100 m a.s.l., in order to differentiate mid- and high altitudes, for the Vercors massif and the Northern and Southern Alps. Results for additional massifs are presented in the Supplementary Information (Figs. S1-S207). The two RCM grid points neighbour selection techniques and the three different learning periods presented in Sect. 2.4 were tested. The impact of using 6 h input RCM data instead of
- daily data was also tested, but yielded similar results (not shown). Only results based on daily input are presented because GCM/RCM outputs are often available at this time step on data distribution platforms such as the one of EURO-CORDEX.

The following evaluation criteria were used for temperature, total precipitation and snow depth:

- the seasonal average time series from 1979 to 2010,
- the mean annual cycle over 2 distinct periods : 1980-1994 and 1995-2010,
  - the mean altitudinal gradient over 1979-2010 (determined only for each massif),
  - the correlation and the ratio of standard deviations between RCM time series and SAFRAN for each variable and as a function of the integration time (from 1 day to several years) from 1979 to 2010,
- the cumulated probability density function (PDF) of daily variables over 1979-2010,
  - the root mean square error (RMSE) and the mean bias over 1979-2010, computed over seasonal integration periods,
- scores specific to the detection of precipitation events over 1979-2010 : the probability of detection (POD = n<sub>hh</sub>/(n<sub>hh</sub> + n<sub>hd</sub>)), the false alarm rate (FAR = n<sub>dh</sub>/(n<sub>dh</sub> + n<sub>hh</sub>)), the probability of false detection (POFD = n<sub>dh</sub>/(n<sub>dh</sub> + n<sub>dd</sub>)) and the true skill score (TSS = POD FAR), where n<sub>hh</sub> is the number of days which are wet in the reanalysis and wet in the adjusted RCM, n<sub>dd</sub> the number of days which are dry in the reanalysis and dry in the adjusted RCM, n<sub>hd</sub> the number of days which are wet in the reanalysis but dry in the adjusted RCM and n<sub>dh</sub>