# Peer review of "The downscaling and adjustment method ADAMONT v1.0 for climate projections in mountainous regions applicable to energy balance land surface models"

_Geoscientific Model Development, 2016_

## Referee Comment (RC1) · Anonymous Referee #1 · 3 Nov 2016

Review of

The downscaling and adjustment method ADAMONT v1.0 for climate projections in mountainous regions applicable to energy balance land surface models By D. Verfaillie, M. Déqué, S. Morin, and M. Lafayesse Geosci. Model Dev. Discuss., doi:10.5194/gmd-2016-168, 2016

General remark

The present work is potentially relevant, representing a method for adjusting RCM output to the conditions in mountainous environments. I particularly like the approach

to show the immediate consequences of the method's performance (with respect to the individual meteorological parameters) for energy balance-based land surface models, as done in this study by evaluating snow depth results obtained by driving CROCUS with output by ADAMONT.

Besides, I have several major concerns. The reanalysis data set the authors use as a reference is very specific, as it includes average conditions within different mountain massifs for different altitudes. As such, it is hard to see the relevance of this study in a broader context, for example, when focusing on applications that use more common observation data sets (e.g., local-scale observations, or observations on spatially regular grids). With this regard, the terminology "downscaling method" may also be inappropriate (see my comments below). The authors need to discuss potential implications of using their method for observational data sets other than the SAFRAN. I also have important concerns regarding the evaluation of the results. The authors do not show the performance of the method based on independent data. This affects the entire discussion and the conclusions. Also, the evaluation is not performed at the scale of the application (individual massifs), but at a larger scale (Northern Alps, Southern Alps). However, for the application in energy balance based land surface models we are interested in the skill of the method of reproducing more local-scale conditions. I recommend considering the study for publication in Geoscientific Model Development if the authors perform a major revision.

Major comments

1. The authors do not perform an evaluation based on independent data. They use different learning periods, but as far as I can follow the validation is based on all the available data (thus, including the data used in the training). I acknowledge the importance of considering different learning periods, but in each case, the validation should be based exclusively on data which have not been involved in the learning process. This point affects the entire discussion section and the conclusions (incl. the abstract) of the submitted manuscript. See also my specific comments below.

2. Grid point selection: The role of the SAFRAN massifs' centroids location and elevation is not clear. Apparently, using one location to represent an entire massif represents a simplification. Also, a potential altitudinal bias between the centroids' altitudes and the altitudes of the RCM grid point is not representative for the elevation differences apparent in reality. Not exactly knowing the SAFRAN reanalysis, and the definition and importance of the centroids, it is hard to understand why RCM grid points should be more realistic if they correspond in altitude and location to the SAFRAN centroids of the massifs, given that SAFRAN represent a simplification per se, assuming horizontally homogenous conditions for the entire massifs (thus areas ranging from 500 to 2000 km^2). Overall, it is not recommended in statistical downscaling to use single grid points by atmospheric numerical models as predictors, because single grid point data are affected by numerical noise (see also: the concept of optimum scale, or effective resolution of an atmospheric numerical model). Also, data by at the RCM surface may be outperformed by the respective data extracted from the relevant pressure levels (see, e.g., Räisänen and Ylhäisi, 2011, Hofer et al, 2012, and references therein).

3. The term "downscaling method" for the ALADIN procedure is somehow misleading. The RCM has a horizontal grid point distance of 12.5 km^2, thus the individual grid cells cover areas of approx.. 150 km^2, while the SAFRAN assume spatially homogenous conditions within each 300 m altitude band of each massif, which in turn can cover up to 2000 km^2. To my understanding, it is thus not possible to define a horizontal resolution of the SAFRAN reanalysis, in traditional terms. SAFRAN may represent more realistic conditions, in particular regarding the altitudinal differences within each massif, than the RCM. However, "downscaling RCM data to SAFRAN" is certainly not what is intended by the term "downscaling" used for inferring higher-resolution information by coarser-scale atmospheric numerical models in the scientific community. Please clearly discuss the practical differences between fitting a RCM to the SAFRAN reanalysis vs. downscaling a RCM to, e.g., to higher resolution gridded observations. I don't know any reanalysis data set comparable to the SAFRAN reanalysis. I recommend to discuss the implications of using ADAMONT based on other observational data sets as

are more commonly available. Otherwise this study is important only for a very narrow range of applications (i.e., applications based on the SAFRAN data set).

4. The discussion of the results is too lengthy. I see the importance of showing various evaluation criteria. However, the information should be compressed and presented in a more synthesized manner. Also, there are too many figures and the figure fonts are too small. Try to highlight the important points using figures which summarize the results in a more transparent way.

5. The article needs to be proofread by a native English scientist. The language needs improvement.

Specific remarks

Lines 175-178: This sentence is not clear. At this point, the reader does not know what the authors mean with "adjusted" RCM. Particularly, what do you mean with "there is no risk of introducing any artificial inflation of the simulated series"?

Lines 185-186: This sentence implies an assumption. The sensitivity of quantile mapping to circulation may change in different climates. Please clearly distinguish the terms "weather regimes" and "climate".

Eq. 1: Indicate the value of N you used. How was N determined?

Lines 199-200: Provide more information about the clustering method you applied, since it is a crucial step in the downscaling procedure. For example, why exactly four weather regimes?

Line 205: What does the definition of the snow year matter for the downscaling procedure at this point?

Lines 208-213: This step is not clear. Is the selection of SAFRAN dates for each RCM date unique? How is this step automatized? Do you consider autocorrelation in the SAFRAN time series to avoid artificial jumps? Thus this step imply a reordering of

the daily RCM time series in order to best correspond to the temporal ordering in the SAFRAN data?

Line 214 and below: The differences between the daily and subdaily ALADIN values should be as small as possible, how is the relation between the SAFRAN subdaily and the ALADING subdaily values? You may provide a formula which describes the transfer. The equations provided concern only air temperature, not the other variables, and it is not clear in which way the SAFRAN sub-daily values are considered.

Line 246: This step is confusing. So you put two quantile mappings on top of each other?

Line 258: Please clearly define what you evaluate. As far as I can follow, you want to evaluate the output of ADAMONT. However, you repeatedly mix "output of ADAMOMT" with the terms "RCM" and" adjusted RCM". For example, line 286: "ratio of the standard deviations between the RCM time series and SAFRAN". Do you really mean standard deviation of the RCM? Then this criterion is not indicative for the performance of ADA-MONT, but for the performance of the selected RCM grid point without any adjustment. Further, whenever you use "adjusted RCM" I am not sure if you mean the ADAMONT output or some intermediate step in the downscaling procedure. If you evaluate the hourly output by ADAMONT applied to ALADIN, please say so. All over the manuscript: don't use brackets inside brackets (e.g., in the references, line 265-266)

Line 270: So the method consists of "downscaling" values at a massif scale (with downscaling not necessarily being the appropriate term), but then the evaluation is not performed at the massifs' scale, but at a much larger scale. The evaluation needs to be applied at the scale of interest, in this case, the individual SAFRAN massifs. Then, the resulting (numerous) scores may be synthesized (e.g., box plots of scores resulting for individual massifs for the Northern alps, box plots of scores resulting for the individual massifs for the Southern alps). The same for the altitudinal ranges. There are various ways how to summarize and appropriately illustrate performance metrics

for numerous cases (here, variables, massifs, altitudinal ranges, and variants of the method in terms of learning period, grid point selection, a posteriori corrections, ...). However it is important that the performance metric is applied to the scale of interest (and: that the reader does not need to interpret too many figures, see also my major comment above).

Line 286: seasonal average time series is not an evaluation criteria per se. Mean annual cycle and mean altitudinal gradient: the same. Please be more specific.

Line 287: RCM time series or ADAMONT time series?

Line 290: not an evaluation criterion.

Line 317: Is "analysis of different massifs" is limited to the application of scores to the average conditions in the Northern and Southern alps, correctly?

Line 320: Following the authors description, their evaluation is never independent from the training data. I.e., for the three different learning periods applied, performance metrics are always calculated including the training data. Case 1: Training period: 1980-1994, Case 2: training period 1995-2010, Case 3: training period 1980 to 2010. Evaluation period is always 1980-2010, thus always includes the training data. The performance metrics should be calculated based on split – sample validation (if parametric properties are validated, e.g., biases, distributions, e.g., Cannon), or cross-validation (if the temporal sequencing is validated, e.g., mean squared errors, ect.). Otherwise, the validation has no evidence.

Line 462: term "neighbor selection" technique is misleading. Use "grid point selection" instead.

Figure 3 is a bit confusing. I see only one SAFRAN centroid being linked with the closest grid points in x, y, and z. Again, I am not sure what exactly you mean with "adjusted RCM". Is this the final output by the ADAMONT procedure based on ALADIN? Please be consistent with the terminology for the output.

Figure 16: It is hard to make any conclusions based on a visual inspection of Figure 16. Plotting the deviations from the modelled against the observed cumulative PDFs could help. The same with Figures 17-19.

Figures 20 – 22: It is hard to distinguish amongst the different lines. Again, it could help to plot deviations of the model results to the SAFRAN values. Still, there are too many figures and the information content should be compressed (e.g., in terms of summary statistics for each model option, e.g., boxplots of skill scores).

References:

Cannon, A. J., 2016: Multivariate bias correction of climate model output: Matching marginal distributions and intervariable dependence structure. Journal of Climate, 29 (19), 7045–7064, doi:10.1175/JCLI-D-15-0679.1, URL http://dx.doi.org/10.1175/JCLI-D-15-0679.1, http://dx.doi.org/10.1175/JCLI-D-15-0679.1.

Hofer, M., B. Marzeion, and T. Mölg, 2012: Comparing the skill of different reanalyses and their ensembles as predictors for daily air temperature on a glaciated mountain (Peru). Climate Dynamics, 39, 1969–1980, doi:10.1007/s00382-012-1501-2.

Räisänen, J., and J. S. Ylhäisi, 2011: How much should climate model output be smoothed in space? Journal of Climate, 24 (3), 867–880, doi:10.1175/2010JCLI3872.1.

---

## Referee Comment (RC2) · Anonymous Referee #2 · 16 Nov 2016

This paper describes a new statistical adjustment method intended to correct the biases in regional climate simulations in order to force land surface models in mountainous regions, and its application over the French Alps. The method is applied to the results of a RCM simulation forced by an atmospheric reanalysis. Precipitation and temperature after correction, and snow cover after land surface modelling with corrected forcing variables are compared to observations.

The paper could be an interesting and useful addition to the field. The adjustment method is sound, and the evaluation work is serious. It may be publishable after major

revisions. However, the description of the method needs to be much improved and the authors need to totally rethink how they present the results of the evaluation, with much less figures, but that better synthesize the results (see my major comment bellow). Moreover, the authors also need to demonstrate that the novelties of the adjustment method (quantile-quantile mapping that depends on large scale circulation; method used for the temporal downscaling from daily to hourly outputs) are useful. I also think that the English is not very good, and need to be improved.

General comments

The paper is not particularly well written (despite visible efforts), with long and awkward sentences that sometimes make the paper difficult to understand.

Some important methodological aspects of the proposed adjustment method are not well described and sometimes not described at all. For example, the basic quantile-quantile mapping algorithm is not described precisely. In the description of the adjustment method, the authors simply describe the different steps very factually, but don't give the precise objective of the step (which is not always obvious) and very seldom justify the proposed solution (see my specific remarks).

The authors have produced a very large number of figures (28 figures with a very large number of sub-figures. In the end, we have hundred of illustrations and even more in supplementary materials). The sub-figures are often very small and therefore difficult (and sometimes impossible) to read. I think it is the job of the authors to do an effort to synthesize their results with a limited number of relevant illustrations, and to only show the important results (at least in the main paper): I disagree with the approach that consists in producing as much as possible illustrations and letting the reader finds what is important.

The core of the adjustment method, quantile-quantile mapping, is well known and has been widely used. An originality of the approach proposed in the paper (even if it is not really the first time it is used, as noted by the authors) is to apply the quantile-quantile

mapping by regime of large-scale circulation. Unfortunately, the authors do not demonstrate the interest of this approach. Is it really useful to do that? A second originality is the method used to obtain hourly data from the adjusted daily RCM output, which is often a necessary step to be able to force a land surface model. The authors propose a quite sophisticated approach, but do not evaluate its interest compared to simpler approaches (e.g. daily cycle from an analogous day without adjustment, or climatological diurnal cycle), neither directly (using observations with hourly resolution, I'm sure that some are available on the study area) nor indirectly (for example by comparing the simulated snow cover obtained with the different approaches). The authors should demonstrate that the novelties they introduce are really useful. It would significantly reinforce the interest of the paper.

Specific remarks

L67-68. I'm not sure to understand. It depends on how one deals with the distribution tail, I think.

L98-102. Unclear and awkward sentence.

L102-104. Not clear

L127-128. OK, but in the end, the adjustment method is intended to correct the output of classical RCM projections such as the ones from Euro-Cordex. A smaller domain likely results in smaller biases compared to the biases found in typical RCM projections. Therefore the evaluation shown in this paper does not demonstrate that the adjustment method is able to deal correctly with the larger biases from classical RCM projections. I think it is a limitation of this work that should be pointed (including in the conclusion).

L130-144. The authors need to explain what exactly is SAFRAN, how the values at different elevations are obtained etc. It may help to better understand some of their choices for the adjustment method. They also talk about the "centroids" of SAFRAN massifs. How are the centroids defined, and what do they represent?

Part 2.3. It would be good to give the forcing variables, their time step etc. in this section.

L165. "Centroids": see a previous remark.

L173. Please provide a more precise description about the exact algorithm used for the quantile-quantile mapping. Only a very brief general idea is given for the moment. For example, how many quantiles are used? How does it work for the values between quantiles: is a linear interpolation is used? How does it work for the values greater than the higher quantile? How the fact that the probability of precipitation in the RCM is different than in SAFRAN is dealt with?

L174. Plotting? I hope that the authors do not really plot the simulated quantiles versus the observed quantiles.

L179. Most adjustment methods make the same hypothesis...

L187-194. For each massif, the authors use a single RCM grid point, the closest (either horizontally or also taking into account the vertical distance) of the massif centroid. Another solution, maybe better, would be to use all the RCM grid points within a massif, and use, based on their altitude, the most appropriate point for each elevation band within the massif. Another possibility, a priori more logical than the single point approach of the authors, would be to average all the RCM grid points within a massif. Obviously, the statistical properties of the spatial average are not the same than for a single point, but the values from SAFRAN on a massif are already spatial averages, right? I think it could make more sense. In any case, the authors need to justify their approach.

L201. Hourly to daily what?

L203-204. What do the authors mean by "each point". Each centroid? Or each elevation band within a massif? If they mean elevation band, using the word "point" is confusing.

L204. I'm not sure to understand why this precision is necessary at this point.

L210. I don't really understand how the "analogous dates" work. The authors need to give the general rationale of their approach, justify the choices they made, and better explain the step. Is there just one analogous date used? Is it just one random date among all the dates that match the different criteria? What is the justification for these criteria? In what sense the date is really "analogous"? The authors could search for a real analogous date, with similar temperature and precipitation over the massif for example. The authors need to explain the rationale behind the use of a day "consistent" in terms of precipitation. And why do they look at the average of precipitation over the Alps and not at the average over the massif of interest? Why the consistency is only defined in terms of occurrence of rain? The intensity does not matter?

L234. How does the optimal value of alpha is chosen precisely? Is it the same at each point?

L255 (point 8 actually). I don't really understand step 8. It seems that, first, total precipitation is adjusted. Then there is a phase separation given temperature and then rainfall and snowfall are re-adjusted separately (only in a variant it seems later in the paper)? Please improve the clarity of the description of this step (rationale and methodology).

L279. I don't see in section 2.4 where the different learning periods are introduced.

L286 "determined only for each massif". I'm not sure to understand.

L349. The "evidenced"? The entire sentence is awkward.

L352. "average altitudinal gradient"? I see the averages for each elevation band in this figure: the gradients are not plotted.

L415. "After 1 month of integration"? This formulation is not very good, I think.

L472. It is really useful to plot hundred of time series (in the main document)? I think

some integrated scores would be much better. Temporal averages in addition to the correlations shown later would be largely sufficient, I think.

L504-505. I don't think that the good scores are mainly due to the adjustment method. The small size of the RCM domain is likely the main responsible for the good correlations. With a small domain, the RCM results are very constrained by the boundary conditions as noted by the authors in a different context. The affirmation is therefore misleading (and references would be needed in any case).

L550-551. Why? The authors do not explain how they deal with extremes values in their algorithm (there are many possibilities. . .). It is therefore difficult for the reader to understand this affirmation.

L564. The temporal transferability is only very partially tested. To my opinion, it is not a major problem that the mean state changes with the learning period. What really matters is whether the trends or the differences between two periods change with the reference period. This is not assessed in the paper, and I think this point should be made.

L630. As noted previously, it does not really make sense to compare the results of different adjustment methods applied to different domains (and RCMs). The differences of performances are more likely to result from the differences of models and domains than from the adjustment methods...

L652-656. As the sentence is written, one may think that the authors want to apply the adjustment method over the entire Europe. Is it really the case? (which data-set would be used instead of SAFRAN in this case?). Or, they simply want to use RCM simulations with a larger domain, as I suspect?

L657. "RCM model" : The M of RCM stands for model.

---

## Referee Comment (RC3) · Anonymous Referee #3 · 19 Nov 2016

General comments

The paper is generally interesting and could provide useful tool to adjust climate data on mountain regions. Especially the adjustment of meteorological variables which affect snow depth is highly welcomed as the accumulation and melting of snow is usually difficult to reproduce even on the areas with relatively constant altitude. Although the paper is promising I have some major comments which I think should be considered before publishing:

1. This paper was hard to read and in some parts to understand as it uses difficult

language and too long sentences.

2. Too many figures. Authors should reduce the amount to half at the actual manuscript and really think through what are the most important figures essential in supplementary material. Despite the authors' good intent the supplementary material with 207 figures is too much. Authors can not assume any reader to have time or willingness to go through those all. It is not good practice to refer to figures 1-207 (!) with every result authors show. 1-3 figures per result should be enough. Font size in figures is too small. It is not stated in every figure caption are the values hourly/daily/monthly/seasonal mean values. "Mean precipitation" does not tell much when the reader is not familiar with the study area and its climatic features.

3. Authors state they have adjusted also wind speed, humidity, and short- and long-wave radiation but do not show any results for these variables. It would have been interesting to see how large effect these variables actually have on snow depth and how much does the bias correction improve the results. This is especially interesting as authors have used hourly data where the variability can be larger than in monthly means.

4. Although the quantile-quantile mapping is well known it is unclear how it is implemented in this study. Especially how is the extreme tail of distributions (>99.5%) handled? Add a short description and clarify the description of ADAMONT method as it is currently hard to follow.

5. Why are RCM's daily values disaggregated to hourly if all results are still presented as daily/monthly/seasonal mean? Authors should make it clear why the hourly data is important for this study.

6. Why are the results shown as mean values for larges areas if the downscaling/adjustment is done separately for each massifs? This smooths especially the extreme values from the data and hides partly the true performance of the method.

7. Be consistent with the names and definitions throughout the paper. Be spearing with acronyms especially if those are used only once.

Specific comments

Lines 67-69: As far as I know the quantile mapping is restricted to the range of observations unless there is added some method to handle the larger values than what was found from the learning period.

Lines 109-112: State clearly that only past climate is studied in this study. I thought also future period was considered here.

Lines 119-120: what about section 4.

Line 165: centroid = center point of some grid point or massif area?

Line 195: What is the range of elevation factor N (0-1, 100-1000 etc.)? How it depends on the altitude?

Lines 201-202 and 214-216: Why is the hourly SAFRAN data first integrated to daily, then used in the quantile mapping function with RCM daily data and then the adjusted RCM daily data is disaggregated to hourly using the same hourly SAFRAN data? Why isn't the RCM data disaggregated to hourly before the ADAMONT adjustment?

Lines 201-202: These integration methods are not clear to me. Have those been shortly explained somewhere? Table 1, method column.

Lines 206-207: Here time period for RCM is 1980-2010 but in figures is used 1979-2010. Why? Describe shortly the quantile mapping method used. There are different variations depending on the treatment of extreme values.

Lines 208-214: I did not understand this step. Is this step 6 supposed to clarify the step 4 or to precede the step 7? Is this step done daily or monthly? And how does it differ from the step 4 where seasonal percentiles were calculated?

253-254: In Olsson et al. (2015) they found that the separation of temperature to dry and wet days produced unrealistic results compared to observations and they used unseparated temperature data for the final results. Was any comparison made with and without separation of precipitation to rain and snow?

Lines 269-271: How much does this grouping decrease and smooth the extreme values?

Lines 277-278 (throughout the paper): It is not a good practice to ask the reader to go through 207 figures to get some clue what would the conclusion of the results be.

Lines 184-301: Why slightly different periods 1979-2010 and 1980-2010 are used in results?

Lines 293-300: Does this mean the relative proportions of wet and dry days are calculated from the whole period separately for RCM and reanalysis and then used to calculate the specific scores or were these calculated so that it had to be dry or wet in both RCM and reanalysis at the same day(hour)? In RCMs the relative proportions should be similar to observations/reanalysis after adjustment but the same weather will probably not occur in the RCM and reanalysis at the same day.

Lines 313-324: Quite long sentence.

Lines: 345-346: Why isn't the table 2 referred already in section 2.5?

Line 349: evidenced=evidences?

Line 354 (throughout the paper): Please be consistent with names, definitions and acronyms. Here "our method" is ADAMONT method?

Lines 358-360: Why is the average precipitation lower in the longest time period compared to the shorter time periods?

Line 367: 9 figures with sub-figures to display the results for RMSE is too much. Please reduce.

Line 371: Why? Is the variability of temperature lower in autumn compared to other seasons?

Lines 373-375: Why is that? Is there less data or too large distances between altitudes?

Lines 381-383: If the figures 7-12 also includes the uncorrected values then it should be stated in their legend.

Lines 385-386: I think bias of 150mm/month sounds quite large. Could you give these over/underestimations as percentage values?

Line 392: Also this bias sounds quite large. See previous comment.

Lines 401-402: Why the N50 degrades results at high altitudes?

Line 409: What does this integration time mean?

Line 425 (throughout the paper): Is there difference between SAFRAN and SAFRAN/Crocus?

Line 277: adjusted RCM = adjusted with ADAMONT method?

Lines 480-482: This is not surprising as the quantile mapping should adjust the learning period values close to observed values! There should be stated how close the ADAMONT methods gets the observational data on the learning period and why there will be greater differences on other periods.

Line 489: What is DSCLIM?

Line 490 (and forward): What is figure 10.1?

Lines 525-528: These acronyms have been already defined in section 2.5.

Lines 546-550: Why are the precipitation underestimated with the longest time period compared to the other periods?

[Figure]

Lines 550-554: How was the extreme tail of distribution handled?

Lines 555-558: Again, the bias correction methods should perform like this and the result is not surprising. How much did these periods differ from each other?

Lines 590-593: ultimate correction = bias correction of rain and snow separately? Please be consistent.

Lines 613-614: No need to explain the acronyms again and again.

Lines 653-667: How does ADAMONT method treat the lowlands where there are no massifs? Gridwise? Table 2: What is the "period considered"? Meaning of RCM APPR?

Figure 18: It is hard to compare the figures as they have different scaling.

Figure 19: The scaling of these figures could be reduced as there is too much white background.

---

## Editor Comment (EC1) · J. C. Hargreaves (Editor) · 28 Nov 2016

All three reviewers see promise in the paper, but only after major revision. The presentation quality must also be significantly improved. So, the manuscript needs a lot of work, but given the general enthusiasm of the reviewers, I will be happy to allow extra time for the revision, should this be required.

---

## Author Comment (AC1) · 27 Jan 2017

**Response to Referee 1**

*We thank R1 for this detailed review, which will enable us to significantly improve our article. Enclosed please find a detailed explanation of the revisions we made based on R1's comments. For your convenience, comments are in bold and our response is in Arial italic. Revisions we made in the manuscript are presented in Arial italic with grey background.*

**General remark**

**The present work is potentially relevant, representing a method for adjusting RCM output to the conditions in mountainous environments. I particularly like the approach to show the immediate consequences of the method's performance (with respect to the individual meteorological parameters) for energy balance-based land surface models, as done in this study by evaluating snow depth results obtained by driving CROCUS with output by ADAMONT.**

**Besides, I have several major concerns. The reanalysis data set the authors use as a reference is very specific, as it includes average conditions within different mountain massifs for different altitudes. As such, it is hard to see the relevance of this study in a broader context, for example, when focusing on applications that use more common observation data sets (e.g., local-scale observations, or observations on spatially regular grids). With this regard, the terminology "downscaling method" may also be inappropriate (see my comments below). The authors need to discuss potential implications of using their method for observational data sets other than the SAFRAN. I also have important concerns regarding the evaluation of the results. The authors do not show the performance of the method based on independent data. This affects the entire discussion and the conclusions. Also, the evaluation is not performed at the scale of the application (individual massifs), but at a larger scale (Northern Alps, Southern Alps). However, for the application in energy balance based land surface models we are interested in the skill of the method of reproducing more local-scale conditions. I recommend considering the study for publication in Geoscientific Model Development if the authors perform a major revision.**

*We thank the reviewer for this review, please see our specific responses to each point below. Concerning the remark « **As such, it is hard to see the relevance of this study in a broader context, for example, when focusing on applications that use more common observation data sets (e.g., local-scale observations, or observations on spatially regular grids).** », we included a sentence to justify the use of SAFRAN in the introduction (l. 99-104) :*

« *However, despite its specificities, SAFRAN is the only reanalysis providing all variables needed to drive energy balance snowpack and land surface models over a long time period (since the 1960s). Moreover, it features a satisfactory altitudinal resolution of 300 m, much more precise than the altitudinal resolution of the RCMs (at a 12.5 km horizontal resolution), which is crucial for assessing the precipitation phase and the altitude variations of snow conditions.* »

*Moreover, we discussed the possible use of other datasets in the conclusion (l. 714-721):*

*« Note that beyond the French mountain regions, the method could be applied in France using the SAFRAN-France gridded reanalysis (Vidal et al., 2010). A Spanish version of SAFRAN was also developed recently (Quintana-Seguí et al., 2016). On a broader scale, the method, although detailed and tested here with the SAFRAN reanalysis as pseudo-observation, could be applied to other observational datasets or meteorological reanalyses, such as ERA-Interim surface fields (Dee et al., 2011) or MESCAN (Soci et al., 2016), providing enough data to drive land surface models. Furthermore, it could be used without application to drive a land surface model for more restricted datasets to simply compute atmospheric diagnostics (such as temperature and precipitation). »*

**Major comments**

**1. The authors do not perform an evaluation based on independent data. They use different learning periods, but as far as I can follow the validation is based on all the available data (thus, including the data used in the training). I acknowledge the importance of considering different learning periods, but in each case, the validation should be based exclusively on data which have not been involved in the learning process. This point affects the entire discussion section and the conclusions (incl. the abstract) of the submitted manuscript. See also my specific comments below.**

*Indeed, we used different learning periods, and we based our validation on different periods too (1980-1995, 1995-2010 and the whole 1980-2010 period, e.g. in Fig. 10), and not only on the 1980-2010 period. Please see our response to R1's specific comment below.*

**2. Grid point selection: The role of the SAFRAN massifs' centroids location and elevation is not clear. Apparently, using one location to represent an entire massif represents a simplification. Also, a potential altitudinal bias between the centroids' altitudes and the altitudes of the RCM grid point is not representative for the elevation differences apparent in reality. Not exactly knowing the SAFRAN reanalysis, and the definition and importance of the centroids, it is hard to understand why RCM grid points should be more realistic if they correspond in altitude and location to the SAFRAN centroids of the massifs, given that SAFRAN represent a simplification per se, assuming horizontally homogenous conditions for the entire massifs (thus areas ranging from 500 to 2000 km^2). Overall, it is not recommended in statistical downscaling to use single grid points by atmospheric numerical models as predictors, because single grid point data are affected by numerical noise (see also: the concept of optimum scale, or effective resolution of an atmospheric numerical model). Also, data by at the RCM surface may be outperformed by the respective data extracted from the relevant pressure levels (see, e.g., Räisänen and Ylhäisi, 2011, Hofer et al, 2012, and references therein).**

*A remark that is shared by all three reviewers is indeed that we did not give enough details about the SAFRAN reanalysis, which is very specific. It is not a traditional gridded reanalysis, but instead the area of interest (the French Alps in our case) is subdivided into different polygons named massifs inside which the meteorological conditions are assumed to be homogeneous. The centre point of each polygon (centroid) plays no specific role in SAFRAN. It was just the way we chose to*

*select RCM grid points close to each massif. The use of single grid points can be justified by the fact that it would not be appropriate to mix (average) RCM pixels with different surface elevations, especially in mountainous regions.*

*We inserted more details about SAFRAN in the manuscript.*

*In the introduction (l. 95-104) :*

*« The SAFRAN meteorological analysis has been developed specifically to address the needs of snowpack numerical simulations in mountainous regions, and contains hourly time series of temperature, precipitation, wind speed, humidity, and short- and longwave radiation for so-called massifs (ranging between 500 and 2,000 km² in the French Alps) by elevation steps of 300 m (Durand et al., 2009a, b). However, despite its specificities, SAFRAN is the only reanalysis providing all variables needed to drive energy balance snowpack and land surface models over a long time period (since the 1960s). Moreover, it features a satisfactory altitudinal resolution of 300 m, much more precise than the altitudinal resolution of the RCMs (at a 12.5 km horizontal resolution), which is crucial for assessing the precipitation phase and the altitude variations of snow conditions. »*

*Section 2.2 has now become Section 2.1 (l. 127-140), and was changed to:*

*« The SAFRAN system is a regional scale meteorological downscaling and surface analysis system (Durand et al., 1993), which provides hourly data of temperature, precipitation amount and phase, specific humidity, wind speed, and shortwave and longwave radiation for each mountain region (or « massif ») in the French Alps (23 massifs, as illustrated in Fig. 1). Unlike traditional reanalyses, SAFRAN does not operate on a grid, but on French mountain regions subdivided into different polygons known as massifs. Massifs (Durand et al., 1993, 1999) correspond to regions ranging approximately between 500 and 2,000 km²  for which meteorological conditions are assumed to be spatially homogeneous but vary with altitude. SAFRAN data are available for elevation bands with a resolution of 300 m. SAFRAN was used by Durand et al. (2009b) to create a meteorological reanalysis over the French Alps by combining the ERA-40 reanalysis (Uppala et al., 2005) with various meteorological observations including in situ mountain stations, radiosondes and satellite data. It was complemented after the end of the ERA-40 reanalysis (2002) by large-scale meteorological fields from the ARPEGE analysis, so that it now spans the period from 1959 to 2016, making it one of the longest meteorological reanalyses available in the French mountain regions. »*

**3. The term "downscaling method" for the ALADIN procedure is somehow misleading. The RCM has a horizontal grid point distance of 12.5 km^2, thus the individual grid cells cover areas of approx.. 150 km^2, while the SAFRAN assume spatially homogenous conditions within each 300 m altitude band of each massif, which in turn can cover up to 2000 km^2. To my understanding, it is thus not possible to define a horizontal resolution of the SAFRAN reanalysis, in traditional terms. SAFRAN may represent more realistic conditions, in particular regarding the altitudinal differences within each massif, than the RCM. However, "downscaling RCM data to SAFRAN" is certainly not what is intended by the term "downscaling" used for inferring higher-resolution information by coarser-scale atmospheric numerical models in the scientific community. Please clearly discuss the practical differences between fitting a RCM to the SAFRAN reanalysis vs. downscaling a RCM to, e.g., to higher resolution gridded observations. I don't know any reanalysis data set comparable to the SAFRAN reanalysis. I recommend to discuss the implications of using ADAMONT based on**

**other observational data sets as are more commonly available. Otherwise this study is important only for a very narrow range of applications (i.e., applications based on the SAFRAN data set).**

*Indeed, the term « downscaling » is not meant in its traditional sense in our study, because we don't perform any horizontal downscaling of RCMs, as R1 rightly points out. It consists more in some sort of « altitudinal downscaling » in mountainous regions and a statistical adjustment, as we produce adjusted RCM data for every 300m elevation band (and in every SAFRAN massif), starting from only one data value at the surface elevation of each RCM grid cell.*

*We decided to change the title of our manuscript accordingly to:*

*« The statistical adjustment method ADAMONT v1.0 for climate projections in mountainous regions applicable to energy balance land surface models »*

*Furthermore, we removed most references to the word « downscaling » and replaced it by the words « statistical adjustment ».*

*We inserted some discussion about the differences between our method and traditional downscaling and discussed the implications of using the ADAMONT method based on other datasets such as gridded reanalyses or observations:*

*l. 99-104 :*

*« (…) However, despite its specificities, SAFRAN is the only reanalysis providing all variables needed to drive energy balance snowpack and land surface models over a long time period (since the 1960s). Moreover, it features a satisfactory altitudinal resolution of 300 m, much more precise than the altitudinal resolution of the RCMs (at a 12.5 km horizontal resolution), which is crucial for assessing the precipitation phase and the altitude variations of snow conditions. »*

*and l. 714-721:*

*« Note that beyond the French mountain regions, the method could be applied in France using the SAFRAN-France gridded reanalysis (Vidal et al., 2010). A Spanish version of SAFRAN was also developed recently (Quintana-Seguí et al., 2016). On a broader scale, the method, although detailed and tested here with the SAFRAN reanalysis as pseudo-observation, could be applied to other observational datasets or meteorological reanalyses, such as ERA-Interim surface fields (Dee et al., 2011) or MESCAN (Soci et al., 2016), providing enough data to drive land surface models. Furthermore, it could be used without application to drive a land surface model for more restricted datasets to simply compute atmospheric diagnostics (such as temperature and precipitation). »*

**4. The discussion of the results is too lengthy. I see the importance of showing various evaluation criteria. However, the information should be compressed and presented in a more synthesized manner. Also, there are too many figures and the figure fonts are too small. Try to highlight the important points using figures which summarize the results in a more transparent way.**

*This is a point that was shared by all three reviewers. We decided to remove figures concerning the Northern and Southern Alps, to keep only figures showing results for the Vercors massif as an example (with larger fonts and better quality) + the same figures for every massif in the French*

*Alps in the Supplement. In the main article, we now have 15 figures instead of 28. Moreover, we decided to include a new synthetic table (Table 3) showing different features (mean values, biases, RMSE values and correlations) for variables of temperature, precipitation and snow depth for every massif in the French Alps + the Northern and Southern Alps, for the « RCM L. 1980-2010 » simulation configuration, at 1200 and 2100 m.*

**5. The article needs to be proofread by a native English scientist. The language needs improvement.**

*We sent our article to a professional English translator who helped improve the langage.*

**Specific remarks**

**Lines 175-178: This sentence is not clear. At this point, the reader does not know what the authors mean with "adjusted" RCM. Particularly, what do you mean with "there is no risk of introducing any artificial inflation of the simulated series"?**

*R1 is right. We removed the term « adjusted » in front of RCM, and inserted the information about resolution of the RCM and SAFRAN in the sentence. The end of the sentence ("there is no risk of introducing any artificial inflation of the simulated series") refers to a potential problem of quantile mapping that can occur when it is applied using observations with a much higher resolution than the RCM, as pointed out by Maraun, 2013 : « If, however, the bias correction also attempts to downscale [i.e., if the correction is against station (or very-high-resolution gridded) data], deterministic variance correction and quantile mapping approaches are not feasible. In general, the spatiotemporal variability at the gridbox scale is much smoother than at the local scale. Yet as only the marginals are corrected and no additional local-scale variability is generated, the temporal dependence and the spatial dependence between locations across grid boxes are those of the gridbox scale. Even more, since the correction is a deterministic mapping, within a grid box the spatial dependence between locations is fully deterministic. Hence, in this downscaling setting also deterministic variance correction and quantile mapping rescale the simulated time series in an attempt to explain unexplained small-scale variability. In other words, they inflate the simulated time series. »*

*We changed our sentence (l. 174-177) to :*

*« Because the spatial resolution of the RCM (12.5 km) is higher than the resolution of the observations (SAFRAN massifs, 500 to 2,000 km²), no spatial downscaling is attempted, so there is no risk of introducing any artificial variance inflation of the simulated series (Maraun, 2013), and therefore quantile mapping is adapted. »*

**Lines 185-186: This sentence implies an assumption. The sensitivity of quantile mapping to circulation may change in different climates. Please clearly distinguish the terms "weather regimes" and "climate".**

*Yes, this is what we imply. With climate change, the frequency of weather regimes may change. Moreover, the model errors are different in different regimes.*

*We introduced more explanation in this paragraph (l. 180-186):*

*« Moreover, Driouech et al. (2009) showed that for mid-latitude climates, such as that in Morocco, quantile mapping adjustment can vary for different weather regimes, because model biases vary in*

*different regimes. Similarly, Addor et al. (2016) demonstrated the sensitivity of quantile mapping adjustment to circulation biases over the Alpine domain. Additionally, the frequency of weather regimes may change in a changing climate (Boé et al., 2006; Cattiaux et al., 2013). To improve the stationarity of our method in a changing climate, weather regimes are thus taken into account in our method. »*

**Eq. 1: Indicate the value of N you used. How was N determined?**

*We tested values of N of 50 and 100, and only showed results for N=50, as explained later in the document. This was rather empirical : using N=50 yielded satisfying neighbouring grid points, while N=100 yielded neighbours that were sometimes too far (e.g., more than 80km in the case of Mont-Blanc for high altitudes) from the SAFRAN centroids.*

*We inserted the values of N we used in this equation (l. 196-198)) :*

*« (…) and N is referred to as the elevation factor. Values of 50 and 100 were tested, but only results using a value of 50 (N50) will be shown in this study. »*

**Lines 199-200: Provide more information about the clustering method you applied, since it is a crucial step in the downscaling procedure. For example, why exactly four weather regimes?**

*More details were provided in step 2 (l. 201-211):*

*« 2. Four different daily weather regimes were diagnosed from ERA-Interim for each season (DJF, MAM, JJA, SON), based on the geopotential height at 500 hPa, following Michelangeli et al. (1995), similar to the method described in Driouech et al. (2010). In the latter studies, only regimes for the winter season are defined. We chose to apply the same method to determine weather regimes for the other seasons as well. The clustering method used is the dynamic cluster method, whose goal is to "find a partition P of the data points into k clusters C1 , C2 ,..., Ck that minimizes the sum of variances (W(P)) within clusters, [...] (by defining) iterative partitions P (n) for which W(P (n) ) decreases with n and eventually converges to a local minimum of W" (Michelangeli et al., 1995). A classifiability and reproducibility analysis in Michelangeli et al. (1995) suggested that 4 weather regimes (k=4) can reasonably be chosen for Europe. This number also ensures a sufficiently large size of the datasets for quantile mapping. »*

*Four weather regimes were chosen for each season, based on previous analyses by Michelangeli et al., 1995. Moreover, we shouldn't calculate more regimes, as it would result in statistically too small datasets, unsuitable for quantile mapping adjustment.*

**Line 205: What does the definition of the snow year matter for the downscaling procedure at this point?**

*R1 is right. In fact, the snow year is not introduced at this point, but at the end of the procedure.*

*We removed this sentence and introduced a last step (step 9, l. 295-298) to explain the resulting time series we obtain :*

*« 9. The resulting ADAMONT-adjusted hourly time series for each variable are obtained for each snow year (from the 1 st August to the 31 st July of the following year), matching the format of the SAFRAN dataset. This makes them easy to use as input of an energy balance land surface model such as SURFEX/ISBA-Crocus. »*

**Lines 208-213: This step is not clear. Is the selection of SAFRAN dates for each RCM date unique? How is this step automatized? Do you consider autocorrelation in the SAFRAN time**

**series to avoid artificial jumps? Thus this step imply a reordering of the daily RCM time series in order to best correspond to the temporal ordering in the SAFRAN data?**

*Indeed, this step was not clear enough. We improved its explanation in the new version of the manuscript.*

*Yes, the selection of SAFRAN dates for each RCM date is unique. For each RCM date, we perform a random draw amongst all available SAFRAN dates, and then browse through the dates chronologically until one meets all the requirements.*

*No, we don't consider autocorrelation in the SAFRAN time series, but we try as much as possible to use consecutive analogous days to avoid artificial jumps in step 7.*

*No, this step does not imply a reordering of the daily RCM time series in order to best correspond to the temporal ordering in the SAFRAN data. Analogous dates in the SAFRAN dataset are only used (in step 7) to reconstruct the daily cycle of the RCM based on hourly time series of the SAFRAN analogue in order to obtain final hourly adjusted RCM time series.*

*This step was re-written (l.229-238) :*

*« 6. For each day in the RCM dataset, an analogous date is chosen in the SAFRAN dataset, matching the following criteria: the month and the regime must be the same as in the RCM dataset, mean precipitation over the Alps must be consistent between datasets to ensure intermediate-scale (accross the French Alps) climatological consistency (i.e. if precipitation in the adjusted dataset is less than a threshold of 1 kg m −2 day −1 , precipitation in the SAFRAN analogue must also be less than this threshold), and whenever possible, consecutive time slices are chosen in the SAFRAN dataset in order to avoid artificial jumps in the final data linked to the choice of analogues. For each RCM date, a random draw amongst all available SAFRAN dates is performed, then the dates are browsed through until one meets all the requirements. This analogous day is then used in step 7 for all variables. »*

**Line 214 and below: The differences between the daily and subdaily ALADIN values should be as small as possible, how is the relation between the SAFRAN subdaily and the ALADING subdaily values? You may provide a formula which describes the transfer. The equations provided concern only air temperature, not the other variables, and it is not clear in which way the SAFRAN subdaily values are considered.**

*We thank R1 for this remark. We have now introduced more details about the procedure, especially for variables other than temperature.*

*Equation 3 was written at the beginning of the step (which changed the order of eqs 2 to 3) and generalized to any variable to explain the relation between the SAFRAN subdaily and the ALADING subdaily values. More details about the procedure for variables other than temperature were also added (l. 239-277):*

*« 7. The adjusted RCM dataset is then disaggregated from a daily integration period into an hourly time step, necessary for driving impact models such as the SURFEX/ISBA-Crocus model, by using the hourly SAFRAN data from each analogous date chosen in the previous step to reconstruct the daily cycle of the data. Hourly adjusted RCM value of any variable can be expressed as a function of the (hourly) SAFRAN value for this variable, as:*

$$X^h_{RCM}(i) = a \times X^h_{SAF} + b, \tag{2}$$

*where $X^h_{RCM}(i)$ is the hourly adjusted RCM value of the variable X and $X^h_{SAF}$ is the hourly SAFRAN*

*value of the same variable from the chosen analogous date (step 6).Different criteria are chosen to calculate a and b, depending on the variable considered (Table 1). For the disaggregation of RCM adjusted temperature from daily to hourly, (…). Sensitivity tests yielded an optimal value of 2 for α. Following eq. (2), eq (3) transforms into : (…).*

*This procedure is only applied for temperature, because the use of the maximum and minimum criterion can lead to important jumps between consecutive days, which is not the case for other variables (Table 1). For humidity, eq. (2) is solved using b=0 and a = $X^{d,adj}_{RCM}(i)$ / $X^h_{SAF}(24h,i)$, so that the hourly adjusted RCM value and the hourly SAFRAN value at the last time step of day i ($X^h_{SAF}(24h,i)$) are equal. For wind speed, the same calculation as for humidity is applied, except if a > 1 (i.e., $X^{d,adj}_{RCM}(i) > X^h_{SAF}(24h,i)$) : then b=$X^{d,adj}_{RCM}(i) - X^h_{SAF}(24h,i)$ is calculated. For humidity and wind speed, if $X^h_{SAF}(24h,i) \leq 10^{-10}$, a=0. For precipitation and radiation, b=0 and a= $X^{d,adj}_{RCM}(i)$ / $X^h_{SAF}(mean,i)$, so that the mean hourly adjusted RCM value and the mean hourly SAFRAN value of day i are equal. For solar radiation, if $X^h_{SAF}(mean,i) \leq 10^{-10}$, a=0. For precipitation, if this is the case, a=1. »*

**Line 246: This step is confusing. So you put two quantile mappings on top of each other?**

*Yes, but the second quantile mapping is only applied to rainfall and snowfall separately, once they are calculated from the hourly total precipitation and temperature and cumulated on a daily basis.*

*The description of step 8 (l. 278-294) was improved and better justified:*

*« Finally, total precipitation is separated into rainfall and snowfall based on hourly adjusted temperature (a threshold of 1 °C is used for the transition from snow to rain, consistent with the approach used in SAFRAN). As mentioned above, inter-variable consistency is not guaranteed by quantile mapping. Consistency between temperature and precipitation is the most critical in this study, because we focus on mountain regions where snow plays an important role. As precipitation and temperature were corrected independently from each other (step 5), and because the adjustment can differ for the different precipitation phases, the relationship between temperature and precipitation phase may be modified by quantile mapping, so that the adjusted rain and snow distributions may lose consistency. To avoid this, Olsson et al. (2015) separate their temperature data into wet and dry days before adjustment. In our case an additional quantile mapping against SAFRAN is applied for daily cumulated RCM rainfall and snowfall separately. Hourly adjusted RCM rainfall and snowfall ($a_2$) are then determined by applying the ratio between daily rainfall or snowfall (taken separately) after quantile mapping ($A_2$) and daily rainfall or snowfall before quantile mapping ($A_1$) to the hourly rainfall or snowfall before quantile mapping ($a_1$):*

$$a_2 = a_1 \times A_2/A_1 \qquad\qquad (5)$$

*If $A_1 = 0$ and $A_2 = 0$, then $a_2 = 0$. If $A_1 = 0$ and $A_2 = 0$, then $a_2 = A_2$ .»*

**Line 258: Please clearly define what you evaluate. As far as I can follow, you want to evaluate the output of ADAMONT. However, you repeatedly mix "output of ADAMOMT" with the terms "RCM" and" adjusted RCM". For example, line 286: "ratio of the standard deviations between the RCM time series and SAFRAN". Do you really mean standard deviation of the RCM? Then this criterion is not indicative for the performance of ADAMONT, but for the performance of the selected RCM grid point without any adjustment. Further, whenever you use "adjusted RCM" I am not sure if you mean the ADAMONT output or some intermediate step in the downscaling procedure. If you evaluate the hourly output by ADAMONT applied to ALADIN, please say so.**

*Indeed, we thank R1 for this remark. « adjusted RCM » is intended as the ADAMONT output.*

*We tried to be more consistent throughout the manuscript, for example :*

*l. 300-301 :*

*« To evaluate our method, outputs by ADAMONT applied to the Météo France ALADIN RCM forced by ERA-Interim (1980-2010) were analysed »*

*or, l. 333-335:*

*« the correlation and the ratio of standard deviations between time series of ADAMONT applied to the RCM and time series of SAFRAN for each variable and as a function of the integration window (from 1 day to several years) from 1980 to 2010 ; »*

**All over the manuscript: don't use brackets inside brackets (e.g., in the references, line 265-266)**

*This problem was corrected.*

**Line 270: So the method consists of "downscaling" values at a massif scale (with downscaling not necessarily being the appropriate term), but then the evaluation is not performed at the massifs' scale, but at a much larger scale. The evaluation needs to be applied at the scale of interest, in this case, the individual SAFRAN massifs. Then, the resulting (numerous) scores may be synthesized (e.g., box plots of scores resulting for individual massifs for the Northern alps, box plots of scores resulting for the individual massifs for the Southern alps). The same for the altitudinal ranges. There are various ways how to summarize and appropriately illustrate performance metrics for numerous cases (here, variables, massifs, altitudinal ranges, and variants of the method in terms of learning period, grid point selection, a posteriori corrections, . . .). However it is important that the performance metric is applied to the scale of interest (and: that the reader does not need to interpret too many figures, see also my major comment above).**

*The evaluation was performed at the scale of the Vercors massif (one individual massif taken as an example). Moreover, supplementary figures display the same evaluation for every single massif in the French Alps (23 massifs in total). Results integrated at the scale of the Northern and Southern Alps are now only provided in Table 3 and not anymore in several plots. Nevertheless, depending on the application, it can be appropriate to operate at the scale of a given massif or at the scale of a wider geographical area such as Northern and Southern Alps. For example, studies addressing water ressources, natural snow abundance, winter tourism, hydrological regimes etc. do not require the same level of geographical resolution. In addition, most of the climate change signal is regional, so that for most regional-scale assessments model results aggregated for the Northern or Southern Alps, repsectively, will be amply sufficient. Concerning the need to reduce the number of figures, we agree with R1, and this was addressed above (General remark n°4).*

**Line 286: seasonal average time series is not an evaluation criteria per se. Mean annual cycle and mean altitudinal gradient: the same. Please be more specific.**

*We thank R1 for this remark. The sentence (l. 329) was changed to :*

*« The following features were analysed for temperature, total precipitation and snow depth »*

**Line 287: RCM time series or ADAMONT time series?**

*This was already addressed above (R1 specific remark Line 258)*

**Line 290: not an evaluation criterion.**

*Please see our answer above (R1 specific remark Line 286)*

**Line 317: Is "analysis of different massifs" is limited to the application of scores to the average conditions in the Northern and Southern alps, correctly?**

*No. We computed scores for the Vercors massif as an example, and for every massifs in the French Alps in the Supplementary Figures.*

**Line 320: Following the authors description, their evaluation is never independent from the training data. I.e., for the three different learning periods applied, performance metrics are always calculated including the training data. Case 1: Training period: 1980-1994, Case 2: training period 1995-2010, Case 3: training period 1980 to 2010. Evaluation period is always 1980-2010, thus always includes the training data. The performance metrics should be calculated based on split – sample validation (if parametric properties are validated, e.g., biases, distributions, e.g., Cannon), or cross-validation (if the temporal sequencing is validated, e.g., mean squared errors, ect.). Otherwise, the validation has no evidence.**

*Indeed, we used different learning periods, but we did base our validation on different periods, and not only on the whole 1980-2010 period. For example, Section 3.2 Mean seasonal variations : « Fig. 10 represents the mean annual cycle of temperature, precipitation and snow depth for the different ADAMONT-adjusted RCM simulations vs. the SAFRAN/Crocus reanalysis, for the period 1980-1995 and 1995-2010.».*

**Line 462: term "neighbor selection" technique is misleading. Use "grid point selection" instead.**

We corrected this throughout the manuscript.

**Figure 3 is a bit confusing. I see only one SAFRAN centroid being linked with the closest grid points in x, y, and z. Again, I am not sure what exactly you mean with "adjusted RCM". Is this the final output by the ADAMONT procedure based on ALADIN? Please be consistent with the terminology for the output.**

*Yes, this is because we are looking at one specific massif (the Vercors massif, see our comments above), which has one centroid. In the Supp. Figures, this is done for every massif of the French Alps. Concerning the terminology for the output, we corrected this in the new version of the manuscript.*

**Figure 16: It is hard to make any conclusions based on a visual inspection of Figure 16. Plotting the deviations from the modelled against the observed cumulative PDFs could help. The same with Figures 17-19.**

*The conclusion is that the modelled cumulative PDFs are very close to the reanalysis ones, which is reassuring considering the fact that we use a quantile mapping method. We think it is more interesting to show cumulative PDFs than the differences between modelled and observed PDFs. Even if the visual inspection of specific curves may not directly provide estimates of the quantitative deviation between PDFs (there are many quantitative metrics described and used in the manuscript), what is more important here is the fact that the shapes of the PDFs correspond,*

*rather than the deviation for one or several quantile ranges.*

**Figures 20 – 22: It is hard to distinguish amongst the different lines. Again, it could help to plot deviations of the model results to the SAFRAN values. Still, there are too many figures and the information content should be compressed (e.g., in terms of summary statistics for each model option, e.g., boxplots of skill scores).**

*Again, it is hard to distinguish amongst the different lines because the modelled cycle is very close to the observed one, which is a result per se. Showing the shape of the mean annual cycles of temperature, precipitation and snow depth seems again more interesting to us than showing the differences.*

*Concerning the number of figures and the changes we made, please see our responses above (General remark n°4).*

---

## Author Comment (AC2) · 27 Jan 2017

**Response to Referee 2**

*We thank R2 for this detailed review, which will enable us to significantly improve our article. Enclosed please find a detailed explanation of the revisions we made based on R2's comments. For your convenience, comments are in bold and our response is in Arial italic. Revisions we made in the manuscript are presented in Arial italic with grey background.*

**This paper describes a new statistical adjustment method intended to correct the biases in regional climate simulations in order to force land surface models in mountainous regions, and its application over the French Alps. The method is applied to the results of a RCM simulation forced by an atmospheric reanalysis. Precipitation and temperature after correction, and snow cover after land surface modelling with corrected forcing variables are compared to observations. The paper could be an interesting and useful addition to the field. The adjustment method is sound, and the evaluation work is serious. It may be publishable after major revisions. However, the description of the method needs to be much improved and the authors need to totally rethink how they present the results of the evaluation, with much less figures, but that better synthesize the results (see my major comment bellow). Moreover, the authors also need to demonstrate that the novelties of the adjustment method (quantile-quantile mapping that depends on large scale circulation; method used for the temporal downscaling from daily to hourly outputs) are useful. I also think that the English is not very good, and need to be improved.**

*We thank the reviewer for this review, please see our specific responses to each point below.*

**General comments**

**The paper is not particularly well written (despite visible efforts), with long and awkward sentences that sometimes make the paper difficult to understand.**

*We sent our article to a professional English translator who helped improve the langage.*

**Some important methodological aspects of the proposed adjustment method are not well described and sometimes not described at all. For example, the basic quantile-quantile mapping algorithm is not described precisely. In the description of the adjustment method, the authors simply describe the different steps very factually, but don't give the precise objective of the step (which is not always obvious) and very seldom justify the proposed solution (see my specific remarks).**

*We have improved the description and justification of the method in the new version of the manuscript (please see our specific responses below).*

**The authors have produced a very large number of figures (28 figures with a very large number of sub-figures. In the end, we have hundred of illustrations and even more in**

**supplementary materials). The sub-figures are often very small and therefore difficult (and sometimes impossible) to read. I think it is the job of the authors to do an effort to synthesize their results with a limited number of relevant illustrations, and to only show the important results (at least in the main paper): I disagree with the approach that consists in producing as much as possible illustrations and letting the reader finds what is important.**

*This is a point that was shared by all three reviewers. We decided to remove figures concerning the Northern and Southern Alps, to keep only figures showing results for the Vercors massif as an example (with larger fonts and better quality) + the same figures for every massif in the French Alps in the Supplement. In the main article, we now have 15 figures instead of 28. Moreover, we decided to include a new synthetic table (Table 3) showing different features (mean values, biases, RMSE values and correlations) for variables of temperature, precipitation and snow depth for every massif in the French Alps + the Northern and Southern Alps, for the « RCM L. 1980-2010 » simulation configuration, at 1200 and 2100 m.*

**The core of the adjustment method, quantile-quantile mapping, is well known and has been widely used. An originality of the approach proposed in the paper (even if it is not really the first time it is used, as noted by the authors) is to apply the quantile-quantile mapping by regime of large-scale circulation. Unfortunately, the authors do not demonstrate the interest of this approach. Is it really useful to do that? A second originality is the method used to obtain hourly data from the adjusted daily RCM output, which is often a necessary step to be able to force a land surface model. The authors propose a quite sophisticated approach, but do not evaluate its interest compared to simpler approaches (e.g. daily cycle from an analogous day without adjustment, or climatological diurnal cycle), neither directly (using observations with hourly resolution, I'm sure that some are available on the study area) nor indirectly (for example by comparing the simulated snow cover obtained with the different approaches). The authors should demonstrate that the novelties they introduce are really useful. It would significantly reinforce the interest of the paper.**

*We don't consider our approach to be better than other simpler ones. Moreover, we think it is out of the scope of this paper to perform a comparison of different approaches, which is already done as part of other projects (such as the CORDEX ESD experiment). What was missing throughout the manuscript was clearer justifications for the choices we made regarding the approach, especially concerning the regimes of large-scale circulation. This has now been included :*

*For the weather regimes, l. 180-186 were modified as follows :*

*« Moreover, Driouech et al. (2009) showed that for mid-latitude climates, such as that in Morocco, quantile mapping adjustment can vary for different weather regimes, because model biases vary in different regimes. Similarly, Addor et al. (2016) demonstrated the sensitivity of quantile mapping adjustment to circulation biases over the Alpine domain. Additionally, the frequency of weather regimes may change in a changing climate (Boé et al., 2006; Cattiaux et al., 2013). To improve the stationarity of our method in a changing climate, weather regimes are thus taken into account in our method. »*

*Moreover, we included more details about the weather regimes selection in step 2 (l. 201-211) of Section 2.3 :*

*« 2. Four different daily weather regimes were diagnosed from ERA-Interim for each season (DJF, MAM, JJA, SON), based on the geopotential height at 500 hPa, following Michelangeli et al.*

*(1995), similar to the method described in Driouech et al. (2010). In the latter studies, only regimes for the winter season are defined. We chose to apply the same method to determine weather regimes for the other seasons as well. The clustering method used is the dynamic cluster method, whose goal is to "find a partition P of the data points into k clusters $C_1$ , $C_2$ ,..., $C_k$ that minimizes the sum of variances (W(P)) within clusters, [...] (by defining) iterative partitions P (n) for which W(P (n) ) decreases with n and eventually converges to a local minimum of W" (Michelangeli et al., 1995). A classifiability and reproducibility analysis in Michelangeli et al. (1995) suggested that 4 weather regimes (k=4) can reasonably be chosen for Europe. This number also ensures a sufficiently large size of the datasets for quantile mapping. »*

*It is easy to justify why we did not choose simpler approaches. Concerning the approach of using the daily cycle from an analogous day without adjustment, the main factor for temperature change at a local scale is the large-scale radiation balance, not changes in atmospheric circulation. This main factor cannot be captured by using the analog without adjustment. Moreover, Boé (2007) showed that using analogous days without any adjustment did not enable to represent trends linked to climate change correctly. Furthermore, it is well known that the diurnal cycle of temperature and radiation is highly affected by cloudiness so using a climatological diurnal cycle could definitely not be realistic under many circumstances.*

*Using direct observations with hourly resolution would not be appropriate, because we only have few stations where all variables are available on the long-term (including radiation), for example the Col de Porte site, with « only » 22 years of hourly observations (Morin et al., 2012). Measurements at these stations could not be extrapolated to the entire French Alps, it is thus better to use a reanalysis such as SAFRAN.*

**Specific remarks**

**L67-68. I'm not sure to understand. It depends on how one deals with the distribution tail, I think.**

*Yes, indeed. In our case, we use a constant correction after the last quantile (99.5%), so this statement remains valid. We completed this sentence (l. 67-71) to take into account R2's remark :*

*« Moreover, the adjustment is not strictly restricted to the range of observed values in the reference period, which is the case for example for methods based on analog weather patterns (e.g., Déqué, 2007; Themeßl et al., 2011; Rousselot et al., 2012; Dayon et al., 2015), provided that values based on the lowermost and uppermost quantiles are handled appropriately (Gobiet et al., 2015). »*

**L98-102. Unclear and awkward sentence.**

*We reformulated the sentence (l. 104-107):*

*« Using this reanalysis as a pseudo-observation dataset combined with the strong and efficient quantile mapping adjustement methods in order to drive energy balance snowpack and land surface models is a highly desirable goal making full use of the current capabilities of climate impact assessment tools for mountainous regions.»*

**L102-104. Not clear**

*This sentence (l. 107-110) was also reformulated :*

*« In addition, the use of such methods ensures that the chronology of the RCM, which may be affected by climate change through variations of the seasonality of meteorological conditions, will be maintained in the adjusted climate projections.»*

**L127-128. OK, but in the end, the adjustment method is intended to correct the output of classical RCM projections such as the ones from Euro-Cordex. A smaller domain likely results in smaller biases compared to the biases found in typical RCM projections. Therefore the evaluation shown in this paper does not demonstrate that the adjustment method is able to deal correctly with the larger biases from classical RCM projections. I think it is a limitation of this work that should be pointed (including in the conclusion).**

*The size of the domain should not impact significantly the way RCM biases are corrected : quantile mapping will correct those regardless of their amplitude. However, if we want to evaluate the method in terms of chronology, it is better to choose a small domain so that it is better constrained. Anyhow, the size of the FRB12 domain (2200 x 2200 km) used in this study is comparable to the size of the commonly employed RCM ensembles such as EURO-CORDEX (5000 x 5000 km), so that we don't expect to experience problems linked to chronology when applying the ADAMONT method to RCMs such as the ones of EURO-CORDEX. We included a map of the EURO-CORDEX domain in Fig. 1.*

*We moved this paragraph into Section 2.4 « Method evaluation », and included more details (l. 303-309):*

*« We chose to work on a spatial domain smaller than the one used in EURO-CORDEX (domain covering all of Europe, Fig. 1), in order to evaluate the method and not the output of the RCM itself, especially in terms of chronology. Indeed, the smaller the domain, the more it is constrained (Alexandru et al., 2007). Simulations carried out over a domain centered on France, called FRB12 (Fig. 1) were thus evaluated (Sect. 2.4). The domain size (2,200 x 2,200 km) is on the same order of magnitude as the size of the commonly employed RCM ensembles such as EURO-CORDEX (5,000 x 5,000 km).»*

**L130-144. The authors need to explain what exactly is SAFRAN, how the values at different elevations are obtained etc. It may help to better understand some of their choices for the adjustment method. They also talk about the "centroids" of SAFRAN massifs. How are the centroids defined, and what do they represent?**

*Yes, we thank R2 for this advice. A remark that is shared by all three reviewers is indeed that we did not give enough details about the SAFRAN reanalysis, which is very specific. It is not a traditional gridded reanalysis, but instead the area of interest (the French Alps in our case) is subdivided into different polygons named massifs inside of which the meteorological conditions are assumed to be homogeneous. The centre point of each polygon (centroïd) plays no specific role in SAFRAN.*

*In the introduction (l. 95-104) :*

*« The SAFRAN meteorological analysis has been developed specifically to address the needs of snowpack numerical simulations in mountainous regions, and contains hourly time series of temperature, precipitation, wind speed, humidity, and short- and longwave radiation for so-called massifs (ranging between 500 and 2,000 km² in the French Alps) by elevation steps of 300 m (Durand et al., 2009a, b). However, despite its specificities, SAFRAN is the only reanalysis providing all variables needed to drive energy balance snowpack and land surface models over a*

*long time period (since the 1960s). Moreover, it features a satisfactory altitudinal resolution of 300 m, much more precise than the altitudinal resolution of the RCMs (at a 12.5 km horizontal resolution), which is crucial for assessing the precipitation phase and the altitude variations of snow conditions. »*

*Section 2.2 has now become Section 2.1 (l. 127-140), and was changed to:*

*« The SAFRAN system is a regional scale meteorological downscaling and surface analysis system (Durand et al., 1993), which provides hourly data of temperature, precipitation amount and phase, specific humidity, wind speed, and shortwave and longwave radiation for each mountain region (or « massif ») in the French Alps (23 massifs, as illustrated in Fig. 1). Unlike traditional reanalyses, SAFRAN does not operate on a grid, but on French mountain regions subdivided into different polygons known as massifs. Massifs (Durand et al., 1993, 1999) correspond to regions ranging approximately between 500 and 2,000 km²  for which meteorological conditions are assumed to be spatially homogeneous but vary with altitude. SAFRAN data are available for elevation bands with a resolution of 300 m. SAFRAN was used by Durand et al. (2009b) to create a meteorological reanalysis over the French Alps by combining the ERA-40 reanalysis (Uppala et al., 2005) with various meteorological observations including in situ mountain stations, radiosondes and satellite data. It was complemented after the end of the ERA-40 reanalysis (2002) by large-scale meteorological fields from the ARPEGE analysis, so that it now spans the period from 1959 to 2016, making it one of the longest meteorological reanalyses available in the French mountain regions. »*

**Part 2.3. It would be good to give the forcing variables, their time step etc. in this section.**

*The following lines were added (l. 148-150) :*

*« Similarly to most land surface models, it requires sub-diurnal (ideally hourly) meteorological forcing data including air temperature, humidity, incoming longwave and shortwave radiation, wind speed, as well as rain and snow precipitation. »*

**L165. "Centroids": see a previous remark.**

*This was corrected.*

**L173. Please provide a more precise description about the exact algorithm used for the quantile-quantile mapping. Only a very brief general idea is given for the moment. For example, how many quantiles are used? How does it work for the values between quantiles: is a linear interpolation is used? How does it work for the values greater than the higher quantile? How the fact that the probability of precipitation in the RCM is different than in SAFRAN is dealt with?**

*This description was inserted later in this section, in step 4 & 5 (l. 218-228) :*

*« 4. The quantiles (99 percentiles + 0.5 % and 99.5 % quantiles) of the RCM distribution and the SAFRAN distribution are then calculated at each centre point of each massif and for each elevation band, for each variable, each season (DJF, MAM, JJA, SON) and each of the four weather regimes.*

*5. Quantile mapping is then applied to the entire RCM dataset for the period 1980-2010, taking into account the season and the weather regime. For the values between quantiles, a linear interpolation is used. For RCM values greater than the 99.5 % quantile, a constant adjustment based on the value of this last quantile is applied. For precipitation, it can happen that for low*

*quantiles, the probability of precipitation is lower in the RCM than in SAFRAN (i.e. several null values in the RCM, which can correspond to different positive values in SAFRAN). In this case, a random draw is performed amongst the SAFRAN values within the same quantile.»*

**L174. Plotting? I hope that the authors do not really plot the simulated quantiles versus the observed quantiles.**

*No. We replaced the term « plotting » by « comparing ».*

**L179. Most adjustment methods make the same hypothesis. . .**

*Yes, but it is still a disadvantage of the method worth mentioning.*

**L187-194. For each massif, the authors use a single RCM grid point, the closest (either horizontally or also taking into account the vertical distance) of the massif centroid. Another solution, maybe better, would be to use all the RCM grid points within a massif, and use, based on their altitude, the most appropriate point for each elevation band within the massif. Another possibility, a priori more logical than the single point approach of the authors, would be to average all the RCM grid points within a massif. Obviously, the statistical properties of the spatial average are not the same than for a single point, but the values from SAFRAN on a massif are already spatial averages, right? I think it could make more sense. In any case, the authors need to justify their approach.**

*The first alternative solution R2 proposes is exactly what we do when we select the closest grid point also taking into account the vertical distance (N=50). For one given massif, we may have different RCM grid points selected depending on the elevation band, as in Fig. 3 for the example of the Vercors massif. However, we highlighted a clear degradation of scores for high elevations when using this approach, linked to the scarcity of high altitude grid points in ALADIN compared to SAFRAN, resulting in grid points being selected several tens of kilometers from the centre point of most SAFRAN massifs.*

*The second alternative solution you propose was not considered, because it would mean mixing (averaging) different grid cells with different surface elevation, which we think would be inappropriate.*

**L201. Hourly to daily what?**

*Hourly to daily time resolution. This was corrected.*

**L203-204. What do the authors mean by "each point". Each centroid? Or each elevation band within a massif? If they mean elevation band, using the word "point" is confusing.**

*Indeed. We meant each elevation band inside of each massif…*

*We changed the sentence (see our previous response to R2 specific remark L.173).*

**L204. I'm not sure to understand why this precision is necessary at this point.**

*Yes, this is something that was noted by R1 also. In fact, the snow year is not introduced at this point, but at the end of the procedure.*

*We introduced a last step (step 9, l. 295-298) to explain the resulting time series we obtain :*

*« 9. The resulting ADAMONT-adjusted hourly time series for each variable are obtained for each snow year (from the 1 st August to the 31 st July of the following year), matching the format of the SAFRAN dataset. This makes them easy to use as input of an energy balance land surface model such as SURFEX/ISBA-Crocus. »*

**L210. I don't really understand how the "analogous dates" work. The authors need to give the general rationale of their approach, justify the choices they made, and better explain the step. Is there just one analogous date used? Is it just one random date among all the dates that match the different criteria? What is the justification for these criteria? In what sense the date is really "analogous"? The authors could search for a real analogous date, with similar temperature and precipitation over the massif for example. The authors need to explain the rationale behind the use of a day "consistent" in terms of precipitation. And why do they look at the average of precipitation over the Alps and not at the average over the massif of interest? Why the consistency is only defined in terms of occurrence of rain? The intensity does not matter?**

*Indeed, this step was not clear enough. We improved its explanation in the new version of the manuscript. This step was re-written (l.229-238) :*

*« 6. For each day in the RCM dataset, an analogous date is chosen in the SAFRAN dataset, matching the following criteria: the month and the regime must be the same as in the RCM dataset, mean precipitation over the Alps must be consistent between datasets to ensure intermediate-scale (accross the French Alps) climatological consistency (i.e. if precipitation in the adjusted dataset is less than a threshold of $1 kg m^{-2} day^{-1}$, precipitation in the SAFRAN analogue must also be less than this threshold), and whenever possible, consecutive time slices are chosen in the SAFRAN dataset in order to avoid artificial jumps in the final data linked to the choice of analogues. For each RCM date, a random draw amongst all available SAFRAN dates is performed, then the dates are browsed through until one meets all the requirements. This analogous day is then used in step 7 for all variables. »*

*The preponderant criteria in the choice of analogous date are the month and the weather regimes. The occurrence of precipitation was added to limit the domain for the random draw and mostly to benefit from an hourly chronology of precipitation in the analog. The intensity of precipitation could have been used, but this was not tested, and it is not necessarily a large-scale characteristic.*

**L234. How does the optimal value of alpha is chosen precisely? Is it the same at each point?**

*Yes, alpha stays the same at each point. The optimal value of alpha (2) was chosen empirically, to obtain the best possible balance between the importance of the minimisation of differences between daily and hourly ALADIN minima and maxima and the minimisation of the jump between two consecutive days.*

**L255 (point 8 actually). I don't really understand step 8. It seems that, first, total precipitation is adjusted. Then there is a phase separation given temperature and then rainfall and snowfall are readjusted separately (only in a variant it seems later in the paper)? Please improve the clarity of the description of this step (rationale and methodology).**

*Yes, this is what we do. A method separating rain and snow before adjustment was tested, but it did not yield satisfying results.*

*The description of step 8 (l. 278-294) was improved and better justified:*

*« Finally, total precipitation is separated into rainfall and snowfall based on hourly adjusted temperature (a threshold of 1 °C is used for the transition from snow to rain, consistent with the approach used in SAFRAN). As mentioned above, inter-variable consistency is not guaranteed by quantile mapping. Consistency between temperature and precipitation is the most critical in this study, because we focus on mountain regions where snow plays an important role. As precipitation and temperature were corrected independently from each other (step 5), and because the adjustment can differ for the different precipitation phases, the relationship between temperature and precipitation phase may be modified by quantile mapping, so that the adjusted rain and snow distributions may lose consistency. To avoid this, Olsson et al. (2015) separate their temperature data into wet and dry days before adjustment. In our case an additional quantile mapping against SAFRAN is applied for daily cumulated RCM rainfall and snowfall separately. Hourly adjusted RCM rainfall and snowfall ($a_2$) are then determined by applying the ratio between daily rainfall or snowfall (taken separately) after quantile mapping ($A_2$) and daily rainfall or snowfall before quantile mapping ($A_1$) to the hourly rainfall or snowfall before quantile mapping ($a_1$):*

$$a_2 = a_1 \times A_2/A_1 \hspace{4cm} (5)$$

*If $A_1 = 0$ and $A_2 = 0$, then $a_2 = 0$. If $A_1 = 0$ and $A_2 = 0$, then $a_2 = A_2$ .»*

**L279. I don't see in section 2.4 where the different learning periods are introduced.**

*You're right. They're introduced in Section 2.7 ! The sentence (l. 324-325) was corrected :*

*« The two RCM grid points neighbour selection techniques and the three different learning periods (1980-1995, 1995-2010 and 1980-2010, see Sect. 2.7) were tested. »*

**L286 "determined only for each massif". I'm not sure to understand.**

*We meant we did not calculate the altitudinal gradient for the Northern and Southern Alps (which would make little sense). However, we have now removed figures for the Northern and Southern alps, thus we changed the sentence (l. 332) into :*

*« – the mean value for each elevation band over 1980-2010 ; »*

**L349. The "evidenced"? The entire sentence is awkward.**

*« evidence», this was corrected. The sentence (l. 400-402) was changed to :*

*« This section provides the evidence needed to assess the ability of the ADAMONT method to reproduce the statistical characteristics of SAFRAN for temperature, precipitation and snow depth from daily RCM outputs. »*

**L352. "average altitudinal gradient"? I see the averages for each elevation band in this figure: the gradients are not plotted.**

*This was corrected (l.403-405 and throughout the manuscript):*

*« Figure 3 presents the location of the Vercors massif and its average temperature, precipitation and snow depth values for each elevation band (...) »*

**L415. "After 1 month of integration"? This formulation is not very good, I think.**

*This was corrected (l. 473 and throughout the manuscript):*

*« For integration windows of one month or more, (...) »*

**L472. It is really useful to plot hundred of time series (in the main document)? I think some integrated scores would be much better. Temporal averages in addition to the correlations shown later would be largely sufficient, I think.**

*Please see our previous response concerning figures (3rd R2 General comment).*

**L504-505. I don't think that the good scores are mainly due to the adjustment method. The small size of the RCM domain is likely the main responsible for the good correlations. With a small domain, the RCM results are very constrained by the boundary conditions as noted by the authors in a different context. The affirmation is therefore misleading (and references would be needed in any case).**

*Please see our response to a previous comment on this specific point (R2 Specific comment L127-128). Indeed, the correlations we evaluate are due to the whole system, not only to the ADAMONT method.*

**L550-551. Why? The authors do not explain how they deal with extremes values in their algorithm (there are many possibilities. . .). It is therefore difficult for the reader to understand this affirmation.**

*Yes, R2 is right. We have now included more details about how we deal with the tail of our ditributions in Sect. 2.3 step 5 (see our previous comment to R2 Specific remark L173).*

**L564. The temporal transferability is only very partially tested. To my opinion, it is not a major problem that the mean state changes with the learning period. What really matters is whether the trends or the differences between two periods change with the reference period. This is not assessed in the paper, and I think this point should be made.**

*R2 is right. However, trends in SAFRAN are too uncertain and affected by the heterogeneity of the assimilated data to be evaluated in details (Vidal et al., 2010). At least, the differences between the periods can be deduced from the figures.*

**L630. As noted previously, it does not really make sense to compare the results of different adjustment methods applied to different domains (and RCMs). The differences of performances are more likely to result from the differences of models and domains than from the adjustment methods…**

*In our study, the domain is larger than the ones used in Lafaysse (2011) and Lafaysse et al. (2014), and we obtain similar or even better results. So, if we consider your previous remark that larger domains give larger biases, it means that our method is at least as performant as the ones of Lafaysse (2011) and Lafaysse et al. (2014). Moreover, the latter methods use only reanalyses and statistical downscaling (without using RCMs), thus current RCMs and the ADAMONT method perform at least as well, regardless of the recent evolution of RCM perfomance.*

**L652-656. As the sentence is written, one may think that the authors want to apply the adjustment method over the entire Europe. Is it really the case? (which data-set would be used instead of SAFRAN in this case?). Or, they simply want to use RCM simulations with a larger domain, as I suspect?**

*Yes, the last proposition is correct, we simply want to use RCM simulations with a larger domain (i.e. EURO-CORDEX domain) and apply it to the French mountainous regions (Alps, Pyrenees, Massif Central, Jura, Vosges, Corsica). The sentence (l. 727-730) was re-written :*

*« In the framework of EURO-CORDEX, as we will be working with RCMs driven by GCMs, the objective, on the contrary, will be to focus on a larger RCM domain covering all of Europe, in order to analyse results over the French Alps depending less on the biases of GCMs. »*

**L657. "RCM model" : The M of RCM stands for model.**

*Yes, R2 is right. We removed the term « model ».*

---

## Author Comment (AC3) · 27 Jan 2017

**Response to Referee 3**

*We thank R3 for this detailed review, which will enable us to significantly improve our article. Enclosed please find a detailed explanation of the revisions we made based on R3's comments. For your convenience, comments are in bold and our response is in Arial italic. Revisions we made in the manuscript are presented in Arial italic with grey background.*

**General comments**

**The paper is generally interesting and could provide useful tool to adjust climate data on mountain regions. Especially the adjustment of meteorological variables which affect snow depth is highly welcomed as the accumulation and melting of snow is usually difficult to reproduce even on the areas with relatively constant altitude. Although the paper is promising I have some major comments which I think should be considered before publishing:**

*We thank the reviewer for this review, please see our specific responses to each point below.*

**1. This paper was hard to read and in some parts to understand as it uses difficult language and too long sentences.**

*We sent our article to a professional English translator who helped improve the langage.*

**2. Too many figures. Authors should reduce the amount to half at the actual manuscript and really think through what are the most important figures essential in supplementary material. Despite the authors' good intent the supplementary material with 207 figures is too much. Authors can not assume any reader to have time or willingness to go through those all. It is not good practice to refer to figures 1-207 (!) with every result authors show. 1-3 figures per result should be enough. Font size in figures is too small. It is not stated in every figure caption are the values hourly/daily/monthly/seasonal mean values. "Mean precipitation" does not tell much when the reader is not familiar with the study area and its climatic features.**

*This is a point that was shared by all three reviewers. We decided to remove figures concerning the Northern and Southern Alps, to keep only figures showing results for the Vercors massif as an example (with larger fonts and better quality) + the same figures for every massif in the French Alps in the Supplement. In the main article, we now have 15 figures instead of 28. Moreover, we decided to include a new synthetic table (Table 3) showing different features (mean values, biases, RMSE values and correlations) for variables of temperature, precipitation and snow depth for every massif in the French Alps + the Northern and Southern Alps, for the « RCM L. 1980-2010 » simulation configuration, at 1200 and 2100 m.*

**3. Authors state they have adjusted also wind speed, humidity, and short- and longwave radiation but do not show any results for these variables. It would have been interesting to see how large effect these variables actually have on snow depth and how much does the bias correction improve the results. This is especially interesting as authors have used hourly data where the variability can be larger than in monthly means.**

*Thank you for this remark. Concerning your first statement, we have focussed in this study on variables known as the most important in the study of the snowpack. We think it would be very interesting to look at other variables in a future paper. To see how much the bias correction improves the results would mean using outputs from uncorrected RCMs, which would not be on the same domain, making the direct comparison difficult. Moreover, driving impact models with uncorrected RCMs can lead to inconsistent results.*

**4. Although the quantile-quantile mapping is well known it is unclear how it is implemented in this study. Especially how is the extreme tail of distributions (>99.5%) handled? Add a short description and clarify the description of ADAMONT method as it is currently hard to follow.**

*Indeed, we thank R3 for this remark, which was also shared by the other reviewers. A better description was inserted in step 4 & 5 (l. 218-228) :*

*« 4. The quantiles (99 percentiles + 0.5 % and 99.5 % quantiles) of the RCM distribution and the SAFRAN distribution are then calculated at each centre point of each massif and for each elevation band, for each variable, each season (DJF, MAM, JJA, SON) and each of the four weather regimes.*

*5. Quantile mapping is then applied to the entire RCM dataset for the period 1980-2010, taking into account the season and the weather regime. For the values between quantiles, a linear interpolation is used. For RCM values greater than the 99.5 % quantile, a constant adjustment based on the value of this last quantile is applied. For precipitation, it can happen that for low quantiles, the probability of precipitation is lower in the RCM than in SAFRAN (i.e. several null values in the RCM, which can correspond to different positive values in SAFRAN). In this case, a random draw is performed amongst the SAFRAN values within the same quantile.»*

**5. Why are RCM's daily values disaggregated to hourly if all results are still presented as daily/monthly/seasonal mean? Authors should make it clear why the hourly data is important for this study.**

*Sub-daily data is necessary to drive impact models such as the SURFEX/ISBA-Crocus snow model presented in this study. However, you're right that we haven't emphasized this point enough in the current manuscript. The beginning of step 7 (l. 239-240) was changed to take into account this remark :*

*« The adjusted RCM dataset is then disaggregated from a daily integration period into an hourly time step, necessary for driving impact models such as the SURFEX/ISBA-Crocus model »*

**6. Why are the results shown as mean values for larges areas if the downscaling/adjustment is done separately for each massifs? This smooths especially the extreme values from the data and hides partly the true performance of the method.**

*Results are shown for the Vercors massif as an example, plus for each massif of the French Alps in the Supplement. The choice of showing results for the Northern and Southern Alps can be justified by the spatial scale of the effects of climate change, which is generally regional, and by the use we want to make of this method, i.e. to assess future conditions, whose stakes are generally expressed at the scale of the Northern and Southern Alps. However, as explained in a previous response (R3 General comment n°2), we have decided to show only results for the Vercors massif in the figures (+ for all massifs in the Supplement), and to limit the interpretation to the Northern and Southern Alps in a synthetic table (Table 3).*

**7. Be consistent with the names and definitions throughout the paper. Be spearing with acronyms especially if those are used only once.**

*This was taken into account.*

**Specific comments**

**Lines 67-69: As far as I know the quantile mapping is restricted to the range of observations unless there is added some method to handle the larger values than what was found from the learning period.**

*Yes, we thank R3 for this remark. We have added some explanation to take it into account (and more explanations about how we deal with the distribution tail later in the document) :*

*l. 67-71 :*

*« Moreover, the adjustment is not strictly restricted to the range of observed values in the reference period, which is the case for example for methods based on analog weather patterns (e.g., Déqué, 2007; Themeßl et al., 2011; Rousselot et al., 2012; Dayon et al., 2015), provided that values based on the lowermost and uppermost quantiles are handled appropriately (Gobiet et al., 2015). »*

**Lines 109-112: State clearly that only past climate is studied in this study. I thought also future period was considered here.**

*R3 is right. The method is intended to work also on projections of future climate, but for the evaluation we focussed on recent climate. We adjusted the following sentence (l. 120-122) to state this :*

*« In order to evaluate the performance of the ADAMONT method, here we apply this method to the ALADIN-Climate v5 RCM (Colin et al., 2010) forced by the ERA-Interim reanalysis (Dee et al., 2011) over the period 1980-2010. ».*

**Lines 119-120: what about section 4.**

*Indeed, we forgot to indicate the Discussion. This was added in the new version of the manuscript.*

**Line 165: centroid = center point of some grid point or massif area?**

*Yes, it is the center point of each SAFRAN massif. We have changed the nomenclature in the whole manuscript.*

**Line 195: What is the range of elevation factor N (0-1, 100-1000 etc.)? How it depends on the altitude?**

*This question is related to a remark by R1. We tested values of N of 50 and 100, and only showed results for N=50, as explained later in the document. This was empirical : using N=50 yielded satisfying neighbouring grid points, while N=100 yielded neighbours that were sometimes too far from the SAFRAN centroids (or center points). N does not depend on the altitude, it is simply a factor we use to give more weight to the proximity between grid points and SAFRAN center points in the vertical direction than in the horizontal ones.*

*We inserted the values of N we used in equation (1), l. 196-198:*

*« (…) and N is referred to as the elevation factor. Values of 50 and 100 were tested, but only results with a value of 50 (N50) will be shown in this study. »*

**Lines 201-202 and 214-216: Why is the hourly SAFRAN data first integrated to daily, then used in the quantile mapping function with RCM daily data and then the adjusted RCM daily data is disaggregated to hourly using the same hourly SAFRAN data? Why isn't the RCM data disaggregated to hourly before the ADAMONT adjustment?**

*This was done to keep the consistency of the daily cycle of each variable. Using the analogue technique presented in step 6 to disaggregate the RCM to hourly before the ADAMONT adjustment would give inconsistent results with more frequent discontinuities, because SAFRAN and the original RCMs are too different from each other. Moreover, SAFRAN and other reanalyses projects (UERRA, EURO4M, see for example Soci et al., 2016) are generally more relevant daily, because some of the observations assimilated in the system are only available at a daily time step (for example, precipitation).*

**Lines 201-202: These integration methods are not clear to me. Have those been shortly explained somewhere? Table 1, method column.**

*R3 is right, those methods were not really explained in the manuscript. In fact, the same types of methods are used in step 3 for the integration from hourly to daily and in step 7 for the disaggregation from daily to hourly. R1 asked for a clarification of step 7 regarding those methods. But we should also clarify step 3 accordingly. We added the following short explanation about the integration methods in step 3 (l. 212-217):*

*« The SAFRAN data are integrated from hourly to daily time resolution to match the data content of the available RCM output. The integration method depends on the variable considered (see Table 1) : for temperature, the daily (6 am to 6 am the next day) minimum and maximum values are selected, for wind speed and humidity, the last value of each day (at 6 am) is selected (in order to be comparable to an instantaneous value), and for precipitation and radiation, the daily mean is used. »*

**Lines 206-207: Here time period for RCM is 1980-2010 but in figures is used 1979-2010. Why? Describe shortly the quantile mapping method used. There are different variations depending on the treatment of extreme values.**

*Concerning the period (1979 or 1980 – 2010), we thank R3 for this remark. We forgot to correct the period in the figures and elsewhere in the text, it should be 1980-2010. Moreover, the two half-periods should be 1980-1995 and 1995-2010. This was corrected throughout the manuscript.*

*Concerning quantile mapping, please see our response to general comment n°4 above.*

**Lines 208-214: I did not understand this step. Is this step 6 supposed to clarify the step 4 or to precede the step 7? Is this step done daily or monthly? And how does it differ from the step 4 where seasonal percentiles were calculated?**

*This step precedes step 7, in which SAFRAN hourly series of daily analogues selected for each RCM day in this step are used. It is thus done daily.*

*Indeed, this step was not clear enough. We improved its explanation in the new version of the manuscript (l.229-238) :*

*« 6. For each day in the RCM dataset, an analogous date is chosen in the SAFRAN dataset, matching the following criteria: the month and the regime must be the same as in the RCM dataset, mean precipitation over the Alps must be consistent between datasets to ensure intermediate-scale (accross the French Alps) climatological consistency (i.e. if precipitation in the adjusted dataset is less than a threshold of 1 kg m −2 day −1 , precipitation in the SAFRAN analogue must also be less than this threshold), and whenever possible, consecutive time slices are chosen in the SAFRAN dataset in order to avoid artificial jumps in the final data linked to the choice of analogues. For each RCM date, a random draw amongst all available SAFRAN dates is performed, then the dates are browsed through until one meets all the requirements. This analogous day is then used in step 7 for all variables. »*

**253-254: In Olsson et al. (2015) they found that the separation of temperature to dry and wet days produced unrealistic results compared to observations and they used unseparated temperature data for the final results. Was any comparison made with and without separation of precipitation to rain and snow?**

*Yes, in the results (Figs. 5-6), we use a specific configuration (no corr) where we look at total precipitation (rain + snow) and snow depth without performing the last quantile correction on rain and snow separately. In fact, the separation of precipitation into rainfall and snowfall is an information that needs to be given as input to the SURFEX/ISBA-Crocus snow model, so we decided to produce it anyhow.*

**Lines 269-271: How much does this grouping decrease and smooth the extreme values?**

*We generally look at mean values in the results, not at extreme values. Thus this is out of the scope of this article.*

**Lines 277-278 (throughout the paper): It is not a good practice to ask the reader to go through 207 figures to get some clue what would the conclusion of the results be.**

*The reader is not asked to read all 207 figures, instead he can focus, in case he/she is interested, on the specific massifs he needs. But you're right that we shouldn't put a reference to « Figs S1-S207 » . We removed all these references to keep only « Supplementary Information »*

**Lines 184-301: Why slightly different periods 1979-2010 and 1980-2010 are used in results?**

*Please see our response above (R3 specific remark Lines 206-207).*

**Lines 293-300: Does this mean the relative proportions of wet and dry days are calculated from the whole period separately for RCM and reanalysis and then used to calculate the specific scores or were these calculated so that it had to be dry or wet in both RCM and reanalysis at the same day(hour)? In RCMs the relative proportions should be similar to observations/reanalysis after adjustment but the same weather will probably not occur in the RCM and reanalysis at the same day.**

*We calculated these scores so that it had to be dry or wet in both RCM and SAFRAN at the same hour. However, the ALADIN RCM in the context of this study is driven by the ERA-Interim reanalysis. We are not in the case where a RCM is driven by a GCM, in which case your remark about the same weather not occuring at the same day applies.*

**Lines 313-324: Quite long sentence.**

*Yes indeed. We replaced this long sentence by a list of 4 points (l. 360-374).*

**Lines: 345-346: Why isn't the table 2 referred already in section 2.5?**

*Section 2.6 (and thus Table 2) is now referred to in the following sentence (l. 324-325) in Section 2.4 :*

*The two RCM grid points neighbour selection techniques and the three different learning periods (1980-1995, 1995-2010 and 1980-2010, see Sect. 2.7) were tested.*

**Line 349: evidenced=evidences?**

*Yes, this was corrected.*

**Line 354 (throughout the paper): Please be consistent with names, definitions and acronyms. Here "our method" is ADAMONT method?**

*Yes, this was corrected.*

**Lines 358-360: Why is the average precipitation lower in the longest time period compared to the shorter time periods?**

*This is explained in the Discussion. Please see our response below.*

**Line 367: 9 figures with sub-figures to display the results for RMSE is too much. Please reduce.**

*Please refer to our previous response on this point (R3 General comment n°2).*

**Line 371: Why? Is the variability of temperature lower in autumn compared to other seasons?**

*We have no explanation for this statement.*

**Lines 373-375: Why is that? Is there less data or too large distances between altitudes?**

*This is explained in the Discussion (Section 4.2 Impact of the spatial selection technique)*

**Lines 381-383: If the figures 7-12 also includes the uncorrected values then it should be stated in their legend.**

*This is the reason why we produced Table 2, so that we wouldn't have to explain the different configurations in each figure legend every time.*

**Lines 385-386: I think bias of 150mm/month sounds quite large. Could you give these over/underestimations as percentage values?**

*In Table 3, we have now included the mean annual values of temperature and precipitation, and the winter mean value of snow depth, so that biases can be compared to those mean values.*

**Line 392: Also this bias sounds quite large. See previous comment.**

*Please see our response to the previous comment.*

**Lines 401-402: Why the N50 degrades results at high altitudes?**

*This is explained in the Discussion (Section 4.2 Impact of the spatial selection technique)*

**Line 409: What does this integration time mean?**

*Integration time was meant as an integration window. This was corrected troughout the manuscript.*

**Line 425 (throughout the paper): Is there difference between SAFRAN and SAFRAN/Crocus?**

*SAFRAN is the meteorological reanalysis. SAFRAN/Crocus refers to the Crocus snow model driven by SAFRAN (thus every time snow depth is mentioned). But R3 is right that in this particular sentence, only SAFRAN should be used because we only mention temperature.* This was corrected.

**Line 277: adjusted RCM = adjusted with ADAMONT method?**

*Yes, this was corrected throughout the paper.*

**Lines 480-482: This is not surprising as the quantile mapping should adjust the learning period values close to observed values! There should be stated how close the ADAMONT methods gets the observational data on the learning period and why there will be greater differences on other periods.**

*Thank you for this remark, indeed this is the purpose of quantile mapping.* We included more explanation in Sect. 3.2, where this feature is first noted (l. 527-530):

*« Concerning the choice of learning period, the ADAMONT method performs better during a period which corresponds to its learning period, because quantile mapping adjusts the learning period values close to SAFRAN values, while the adjustment for other periods is based solely on the mapping function determined for the learning period. »*

*In Sect. 3.3 (l. 540-541), we included :*

*« Some significant differences appear when using different learning periods, as already noted in Sect. 3.2,(...) »*

**Line 489: What is DSCLIM?**

*We forgot to introduce it. It's an analog resampling based transfer function algorithm (Pagé et al., 2009).* This was included in the text (l. 551-552).

**Line 490 (and forward): What is figure 10.1?**

*Fig. 10.1 in Lafaysse, 2011 (therein).*

**Lines 525-528: These acronyms have been already defined in section 2.5.**

*Yes, R3 is right.* We removed the explanation of acronyms.

**Lines 546-550: Why are the precipitation underestimated with the longest time period compared to the other periods?**

*The fact that it is underestimated is not meaningful, it could have been overestimated.*

**Lines 550-554: How was the extreme tail of distribution handled?**

*Please refer to our answer above concerning quantile mapping (R3 General comment n°4) .*

**Lines 555-558: Again, the bias correction methods should perform like this and the result is not surprising. How much did these periods differ from each other?**

*We included the following (l. 620-622) :*

*« (…) revealed some significant differences when using different learning periods, linked to the use of quantile mapping, with the ADAMONT-adjusted RCM simulation (...) »*

*The difference between the two sub-periods is explained in Sect. 2.6 (l. 392-394) : « Different learning periods were tested to evaluate their impact: 1980-1995 and 1995-2010, which correspond to periods with contrasting meteorological (and snow) conditions linked to regime shifts (Reid et al., 2015) »*

**Lines 590-593: ultimate correction = bias correction of rain and snow separately? Please be consistent.**

*Yes, this was corrected.*

**Lines 613-614: No need to explain the acronyms again and again.**

*This was removed.*

**Lines 653-667: How does ADAMONT method treat the lowlands where there are no massifs? Gridwise?**

*In its current configuration (using the SAFRAN reanalysis over French massifs), it does not treat them, it is only run over mountain regions where the reanalysis is available. Note however that a France-wide implementation of SAFRAN exists, which uses different zoning than the massif approach in the mountain regions (Vidal et al., 2010).*

**Table 2: What is the "period considered"? Meaning of RCM APPR?**

*The period considered in the figures (1980-2010, 1980-1995 or 1995-2010). This was clarified.*

*RCM APPR is in fact RCM L. (L. stands for learning). We changed the nomenclature and forgot to correct it here. This was corrected.*

**Figure 18: It is hard to compare the figures as they have different scaling.**

*This Figure now corresponds to Figure 8. Indeed, the x-axis scalings are different between 1200m and 2100m subplots, but the point of this Figure is not really to compare the PDF distributions between two elevations, but rather to ensure that for a given elevation, PDFs of ADAMONT-adjusted simulations are close to the SAFRAN one, which is the case.*

**Figure 19: The scaling of these figures could be reduced as there is too much white background.**

*Yes, but we used the same scale for the supplementary figures. Some of them need such a scale.*